# SKIP-IT? THEORETICAL CONDITIONS FOR LAYER SKIPPING IN VISION–LANGUAGE MODELS

## ABSTRACT

Vision–language models (VLMs) achieve incredible performance across a wide range of tasks, but their large size makes inference costly. Recent works have shown that pruning certain VLM layers can improve efficiency with minimal performance loss or even performance improvements. However, these techniques remain underused due to a limited understanding of when pruning is beneficial. In this paper, we **[propose a unified]** framework that characterizes redundancy conditions under which **[pruning can be applied]** to enhance efficiency without sacrificing performance. Our framework proposes experimentally-verifiable and understandable notions of redundancy. Using this framework, we identify that across models, both early and late vision tokens are redundant. We then experimentally verify this finding **[with a case study]** using **[deep insertion]** and early exit layer skipping techniques. The redundancy framework proposed in this paper provides a theoretically-grounded understanding of redundancy in VLMs **[and unifies many of the ideas behind modern layer skipping techniques.]**

## 1 INTRODUCTION

Vision-language models (VLMs) such as BLIP (Li et al., 2022), LLaVA (Liu et al., 2023; 2024a), and more recently, Qwen (Bai et al., 2023; Qwen et al., 2025), Deepseek-VL (Lu et al., 2024), and Gemma (Gemma Team et al., 2025) have become more popular over the past few years due to their impressive performance on Visual-Question Answering (VQA), multimodal reasoning, and image captioning. Nevertheless, these models come with escalating training and inference costs (Jin et al., 2024), creating barriers to widespread adoption. This computational burden stems from how these models typically build upon large language model backbones, requiring the processing of extensive sequences of image tokens alongside text, particularly when handling high-resolution images.

In response to these challenges, techniques to improve VLM efficiency have grown in popularity, aiming to reduce model overhead while maintaining performance. Many approaches build upon efficiency methods from large language models (LLMs), broadly classified as training time or inference time improvements. Training time methods include parameter-efficient approaches such as LoRA (Hu et al., 2022; Dettmers et al., 2023; Biderman et al., 2024) and Mixture of Experts (MoE) (Artetxe et al., 2022; Bao et al., 2022; Lin et al., 2024), while inference time methods encompass token compression, skipping, and quantization. Token compression techniques (Chen et al., 2024a; Vasu et al., 2025; Liu et al., 2025) reduce redundancy in vision representations, while layer skipping, initially developed for efficient LLM inference (Elhoushi et al., 2024), eliminates processing redundancy during forward passes. Recent **[VLM adaptations of layer skipping]** include AdaSkip (He et al., 2025), FlexiDepth (Luo et al., 2025a), and retrained models like DeepInsert (Choraria et al., 2025), MoLe-VLA (Zhang et al., 2025), and $\gamma$-MoD (Luo et al., 2025b). Despite their empirical success, their identification of layers to skip lacks principled justification.

In this work, we propose a learning- and information-theoretic framework to formulate an experimentally easy-to-compute condition that **[unifies the different inference-time layer skipping techniques and]** indicates when there are redundant text and/or vision tokens that can be skipped to improve efficiency without degrading performance. We validate our framework by experimentally verifying the existence of *Functional Redundancy* and *Informational Redundancy*. We then experimentally show the significance of these predictions by **[conducting a case study on Late Entry and Early Exit token pruning]** verifying that the identified redundant layers (and tokens in such

layers) can be skipped without model degradation. **[Conversely,]** we show that model performance degrades when our conditions are not met. This paper aims to showcase a theoretical framework to identify redundancy across VLMs. **[We hope to inspire future token and layer reduction methods by providing these notions of redundancy as a guide for which layers and tokens can be pruned from VLMs]**.

Overall, our contributions are as follows.

1. We propose a theoretical framework to study redundancies between random variables. These redundancies can be used with inter-modal attention analysis to improve inference-time efficiency.

2. We experimentally verify that the necessary conditions from theory are met in the early and late vision tokens across all models. We consider the average cosine distance and the probability of a small cosine distance between adjacent layers.

3. We **[perform a case study using late entry and early exit token pruning]** to validate that fulfilling the redundancy and inter-modal attention conditions corresponds to improved model efficiency with minimal performance degradation, whereas not meeting them leads to performance degradation.

## 2 RELATED WORK

### 2.1 VISION LANGUAGE MODELS

VLMs feed vision and textual tokens into an autoregressive LLM. BLIP models (Li et al., 2022; 2023) were some of the first to introduce vision-language pretraining, with the latter using a Q-former to connect frozen image encoders with LLMs. Flamingo introduced the ability to ingest mixed insertions of images, videos, and text (Alayrac et al., 2022). LLaVA introduced a multimodal instruction-following model. LLaVA-NeXT built on LLaVA with improved reasoning and performance (Liu et al., 2024a). More recently, Molmo improved capabilities in pointing and counting tasks (Deitke et al., 2025). Llama-4 introduced a mixture-of-experts VLM with a massive context length. Specifically, Llama-4 Scout has a token context length of 10M (Meta AI, 2024). In most models, the vision tokens are obtained from a pretrained vision encoder, such as CLIP (Radford et al., 2021) or SigLIP (Zhai et al., 2023). Some core challenges in these models include inference-time inefficiency, multimodal alignment, and vision interpretability.

### 2.2 LAYER SKIPPING **[AND PRUNING]**

Redundancies in tokens and layers have motivated efficiency techniques in both LLMs and VLMs. In LLMs, Shukor & Cord (2024) introduced a method for skipping certain computations (whether entire blocks, specific feed-forward networks or self-attention layers, AdaSkip introduces sublayer-wise skipping for long-context inference (He et al., 2025), while FlexiDepth enables adaptive layer skipping (Luo et al., 2025a). Both exploit redundancy to reduce computation while maintaining performance. Multimodal models have adopted similar strategies: $\gamma$-MoD converts dense layers into sparse Mixture-of-Depth layers (Luo et al., 2025b), MoLe-VLA applies layer skipping to robot manipulation (Zhang et al., 2025), Skip-Vision (Zeng et al., 2025) skips certain feed-forward networks during training and prunes certain key-value pairs during inference, and DeepInsert injects vision tokens at later layers to reduce early-layer overhead (Choraria et al., 2025). Parallel work focuses on token reduction through multimodal skipping or early exit. FastV prunes visual tokens by learning attention patterns (Chen et al., 2024a), Visual Token Withdrawal (VTW) removes visual tokens after certain layers (Lin et al., 2025), PruMerge leverages visual encoder sparsity to discard tokens (Shang et al., 2025), and $ST^3$ prunes redundant vision tokens across layers and dynamically reduces the number of vision tokens across layers. **[Our framework specifically aims to unify these multimodal layer skipping techniques but can possibly be extended to unify inference-time pruning techniques in general (see section 5).]**

## 2.3 VLM INTERPRETABILITY

VLM interpretability focuses on understanding vision–language models, a space that remains less explored than LLM interpretability. Two notable VLM-specific methods are *logit lens* and *causal tracing* (Neo et al., 2025; Jiang et al., 2025; Basu et al., 2024). The *logit lens* uses the unembedding matrix to reveal textual representations of visual tokens and localize visually grounded factual information. *Causal tracing* identifies which layers carry visual knowledge by perturbing prompts to remove image dependence and then copying activations from the unperturbed forward pass. Both techniques adapt earlier LLM interpretability methods: the *logit lens* from Nostalgebraist (2020) and causal tracing from Meng et al. (2023; 2022). Our work also has clear applications to VLM interpretability. The definitions and theorems we propose for measuring redundancies between layers can help determine the functionality of specific layers and give guidance on how information is propagated through the model.

## 3 FRAMEWORK FOR MEASURING REDUNDANCY

### 3.1 DEFINITIONS OF REDUNDANCY

Redundancy, in this setting, quantifies how **[much two random variables (RVs) are "alike". With an application to Transformer architectures in mind, a]** simple and accessible way to capture redundancy is to measure the similarity using an average cosine distance. We refer to this as geometric redundancy.

**Definition 1** (Geometric $\epsilon$-redundancy). *Let $\rho : \mathcal{X} \times \mathcal{X} \to [0, \infty)$ be a symmetric function (e.g. cosine distance or a metric). Then given random variables $X_{\ell-1}, X_\ell$ and $\epsilon > 0$, geometric $\epsilon$-redundancy (or geometric redundancy) is $\mathbb{E}[\rho(X_{\ell-1}, X_\ell)] < \epsilon$.*

We similarly define proximal redundancy as being when two RVs are close together with high probability.

**Definition 2** ($t$-proximal with probability $1 - \epsilon$; Proximal Redundancy). *Let $\rho : \mathcal{X} \times \mathcal{X} \to [0, \infty)$ be a symmetric function (e.g. cosine distance or a metric), $t$ be some threshold value, and $\epsilon > 0$. Then random variables $X_{\ell-1}, X_\ell$ are $t$-proximal with probability $1 - \epsilon$ (or proximally redundant) if $\mathbb{P}[\rho(X_\ell, X_{\ell-1}) < t] > 1 - \epsilon$.*

While geometric and proximal redundancy capture semantic**[/cosine]** similarity, they do not offer much **[interpretation or describe the actual operational redundancy between RVs]**. To rigorously define **[such operational notions of ]** redundancy , we propose these additional **[definitions]**.

**Definition 3** (Functional $\epsilon$-redundancy). *Given a task variable $Z$, two random variables $X_{\ell-1}, X_\ell$, and $\epsilon > 0$, functional $\epsilon$-redundancy (functional redundancy) is $\mathbb{E}[\|E[Z|X_\ell] - \mathbb{E}[Z|X_{\ell-1}]\|_2^2] < \epsilon$.*

**Definition 4** (Informational $\epsilon$-redundancy). *Given random variables $X_{\ell-1}, X_\ell$ and $\epsilon > 0$, informational $\epsilon$-redundancy (informational redundancy) is $H(X_\ell|X_{\ell-1}) < \epsilon$.*

The remainder of this section connects these four notions. Specifically, we show that geometric and proximal redundancy, although less informative definitions, imply more operational forms of redundancy under natural assumptions.

### 3.2 FUNCTIONAL REDUNDANCY

We begin by analyzing functional redundancy, which measures the difference between optimal estimators on a task $Z$.

**Theorem 1.** *Let $X_\ell, X_{\ell-1}$ be unit-norm random variables and $Z$ be the random variable of predictive interest (e.g. normalized hidden representations of layers $\ell, \ell-1$ and the task ground truth respectively). Let $\rho(x, y) = 1 - \frac{\langle x, y \rangle}{\|x\| \|y\|}$. Assume $\mathbb{E}[\rho(X, Y)] < \frac{\epsilon}{2}$ and that*

$$h(x, y) = E[Z|X_\ell = x, X_{\ell-1} = y]$$

*is $\alpha$-Lipschitz in the first argument and $\beta$-Lipschitz in the second. Then $E[\|E[Z|X_\ell] - E[Z|X_{\ell-1}]\|_2^2] < 2(\alpha^2 + \beta^2)\epsilon$.*

*Proof.* We prove this by first translating the cosine distance into mean squared error and then using the optimality of the conditional mean and the tower property. For more details, see Appendix A.2. □

In words, Theorem 1 establishes that under some regularity assumptions, there is a bridge between geometric redundancy and functional redundancy. However, in practice, we can never achieve these optimal estimators, so Theorem 2 bounds empirical estimates of these optimal estimators.

**Theorem 2.** *Let $X_\ell, X_{\ell-1}$ be unit-norm random variables and $Z$ be the random variable of predictive interest, as before. Let $\rho(x,y) = 1 - \frac{\langle x,y \rangle}{\|x\|\|y\|}$ . Assume $\mathbb{E}[\rho(X,Y)] < \frac{\epsilon}{2}$ and that*

$$h(x,y) = E[Z|X_\ell = x, X_{\ell-1} = y]$$

*is $\alpha$-Lipschitz in the first argument and $\beta$-Lipschitz in the second. Let $\hat{f}_\ell$ be a finite-sample estimate of $f_\ell^*(x) = \mathbb{E}[Z|X_\ell = x]$ and $\hat{f}_{\ell-1}(x)$ be a finite-sample estimate of $f_{\ell-1}^*(x) = \mathbb{E}[Z|X_{\ell-1} = x]$. Further let, $\eta_\ell = \mathbb{E}[\|\hat{f}_\ell(X_\ell) - f_\ell^*(X_\ell)\|^2]$ and $\eta_{\ell-1} = \mathbb{E}[\|\hat{f}_{\ell-1}(X_{\ell-1}) - f_{\ell-1}^*(X_{\ell-1})\|^2]$. Then:*

$$\mathbb{E}[\|\hat{f}_\ell(X_\ell) - \hat{f}_{\ell-1}(X_{\ell-1})\|^2] < 3\eta_\ell + 3\eta_{\ell-1} + 6(\alpha^2 + \beta^2)\epsilon.$$

*Proof.* We prove this by again converting from cosine distance to mean squared error and then rewriting $\hat{f}_\ell(X_\ell) - \hat{f}_{\ell-1}(X_{\ell-1})$ as $(\hat{f}_\ell(X_\ell) - f_\ell^*(X_\ell)) + (f_{\ell-1}^*(X_{\ell-1}) - \hat{f}_{\ell-1}(X_{\ell-1})) + (f_\ell^*(X_\ell) - f_{\ell-1}^*(X_{\ell-1}))$. We then upper bound $\hat{f}_\ell(X_\ell) - \hat{f}_{\ell-1}(X_{\ell-1})$ using this new form by invoking Theorem 1 along with the definitions of $\eta_\ell$ and $\eta_{\ell-1}$. For more details, refer to Appendix A.2 □

Thus, under the same regularity assumptions, we obtain guarantees for empirical estimators that mirror those for optimal ones. Importantly, the bound still decays linearly in $\epsilon$, showing that empirical functional redundancy inherits the same behavior. **[Previous layer skipping work has employed similar Lipschitz assumptions on self-attention and feed-forward networks, notably Zeng et al. (2025)]**.

### 3.3 INFORMATIONAL REDUNDANCY

We now turn to informational redundancy, which asks whether **[one RV]** can be (nearly) determined from **[a second RV]**. This is formalized via the conditional entropy $H(X_\ell|X_{\ell-1})$ **[and invokes the continuous Fano's Inequality (Duchi & Wainwright, 2013) and continuous analogs in Braun & Pokutta (2015)]**.

We first get an upper bound on $H(X_\ell|X_{\ell-1})$ from Braun & Pokutta (2015). If this upper bound is sufficiently low, this shows a high statistical correlation between $X_\ell$ and $X_{\ell-1}$. We then use a complementary lower bound from Braun & Pokutta (2015) and Duchi & Wainwright (2013) on the mutual information $I(X_\ell; X_{\ell-1})$. If this lower bound is sufficiently large, we can then show that there is a large amount of information shared between the two RVs, thereby also implying a large amount of redundancy.

To understand these bounds, first define $P_t = \mathbb{P}[\rho(X_\ell, X_{\ell-1}) > t]$ where $\rho$ is some symmetric function (e.g. cosine distance or a metric). $P_t$ is the probability that the "distance" between the two random variables is greater than $t$. Braun & Pokutta (2015) provide an upper bound on $H(X_\ell|X_{\ell-1})$ with respect to $P_t$ while Braun & Pokutta (2015) and Duchi & Wainwright (2013) give a complementary lower bound on $I(X_\ell; X_{\ell-1})$ with respect to $P_t$. The lower bound provided in Duchi & Wainwright (2013) makes some distributional assumptions on $X_\ell$ and gives a more precise, analytical bound while Braun & Pokutta (2015) does not make any distributional assumptions but gives a much looser bound. Refer to Theorems 3, 4, and Corollary 4 in Appendix A.2 for more details on these bounds.

There is substantial prior work, such as Xu et al. (2020); Dissanayake et al. (2025), which has shown that concepts like usable information and unique/redundant information are useful for analyzing machine learning paradigms. See Appendix E for further discussion of the connection to partial information decomposition (PID).

## 3.4 RELATING ALL NOTIONS OF REDUNDANCY

We now give some final results to connect all notions of redundancy.

**Proposition 1.** *Let $\rho : \mathcal{X} \times \mathcal{X} \to \mathbb{R}$ be a symmetric function with $0 \leq \rho \leq 1$. Then*

$$\mathbb{P}[\rho(X,Y) > t] < \frac{\epsilon - t}{1 - t} \quad \text{implies} \quad \mathbb{E}[\rho(X,Y)] < \epsilon,$$

*Proof.* Apply the tail integration formula and split the integral into a part from $0$ to $t$ and another part from $t$ to $1$. Refer to Appendix A.2 for more details. $\qquad\square$

Thus, under natural assumptions, proximal redundancy implies geometric redundancy.

Theorem 5 shows that under some additional natural Markov and boundedness assumptions, informational redundancy implies functional redundancy. **[In particular, since the hidden states are taken after adding the residual, this Markovity assumptions holds.]**

**Theorem 5.** *Suppose there are random variables $Z, X_\ell, X_{\ell-1}$ with $Z \in \mathbb{R}^d$ and $X_\ell, X_{\ell-1}$ continuous **[unit-norm]** random variables. Further suppose $\|Z\|_2 \leq B$ almost surely and that $X_{\ell-1}$— $X_\ell$— $Z$ is a Markov chain. Then $\mathbb{E}[\|\mathbb{E}[Z|X_\ell] - \mathbb{E}[Z|X_{\ell-1}]\|_2^2] \leq 2B^2 I(Z; X_\ell | X_{\ell-1})$.*
*If, in addition, there exists finite $C$ such that $H(X_\ell | Z, X_{\ell-1}) \geq -C$ then $\mathbb{E}[\|\mathbb{E}[Z|X_\ell] - \mathbb{E}[Z|X_{\ell-1}]\|_2^2] \leq 2B^2 (H(X_\ell | X_{\ell-1}) + C)$. In particular, if $X_\ell$ is discrete then $C = 0$ and if $p_{X_\ell | Z, X_{\ell-1}}(x) \leq M \ \forall x$ then $C = \log M$.*

*Proof.* We prove this first by expressing $\mathbb{E}[Z|X_\ell = a] - \mathbb{E}[Z|X_{\ell-1} = b] = \int_{\mathbb{R}^d} z(p_{Z|X_\ell=a}(z) - p_{Z|X_{\ell-1}=b}(z))dz$. We can then bound $\|\mathbb{E}[Z|X_\ell = a] - \mathbb{E}[Z|X_{\ell-1} = b]\|$ using the triangle inequality. We then use Pinsker's inequality to bound $\int_{\mathbb{R}^d} |p_{Z|X_\ell=a}(z) - p_{Z|X_{\ell-1}=b}(z)|dz$ using KL-divergence. Since $X_{\ell-1}$—$X_\ell$—$Z$ is a Markov chain, we can convert this into an upper bound using conditional entropy (Lemma 4). For more details, refer to Appendix A.2. $\qquad\square$

**[Because differential entropy can be negative, one cannot get a direct bound between functional redundancy and informational redundancy in general. However, by considering RVs to be discrete (say, for example, due to quantized activations after compression or due to finite computer memory), then informational redundancy directly implies functional redundancy.]** This result shows that informational redundancy is a more fundamental type of redundancy as it generalizes functional redundancy. Furthermore, the connection to PID gives an interesting way to interpret this bound: under certain conditions, the average difference in performance between optimal MSE estimators based on random variables $X$ and $Y$ is upper bounded by unique information about $X$ that only $X$ has. Refer to Appendix E for more details on PID.

From these results, we have that proximal redundancy implies geometric and informational redundancy, each of which, in turn, implies functional redundancy. Figure 1 summarizes these derived results.

## 4 CASE STUDY ON EARLY EXIT AND LATE ENTRY LAYER SKIPPING

In this section, we **[apply our framework to analyze the case of layer skipping in VLMs for the late entry and early exit pruning cases.]** By combining the framework with additional metrics for quantifying inter-modality interaction via inter-modal attention, we determine layers viable for skipping.

To formally define layer skipping (late entry and early exit), consider a VLM with $n$ layers, $\phi_\theta^n$. Let $X = \begin{pmatrix} X_{text} \\ X_{vis} \end{pmatrix}$ be the input to the VLM. We consider late entry, say of the vision tokens, in the first $\ell$ layers to be $\phi_\theta^{n-\ell}(\begin{pmatrix} \phi_\theta^\ell(X_{text}) \\ X_{vis} \end{pmatrix})$. A visual of this layer skipping can be seen in Figure 2.

Intuitively, from Figure 2, a sufficient condition for late entry is both high levels of redundancy and small vision-to-text attention. This intuition is formalized in the below theorem. **[Formal**

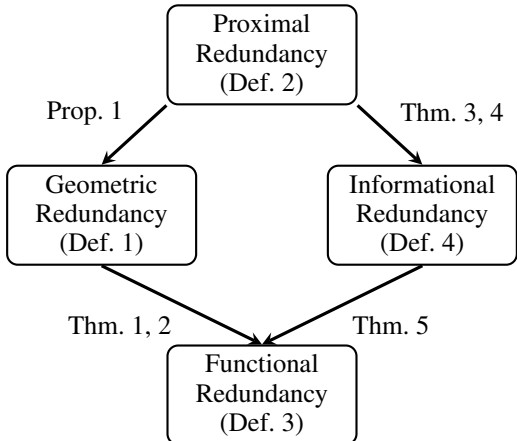

Figure 1: Implication relationships among different notions of redundancy.

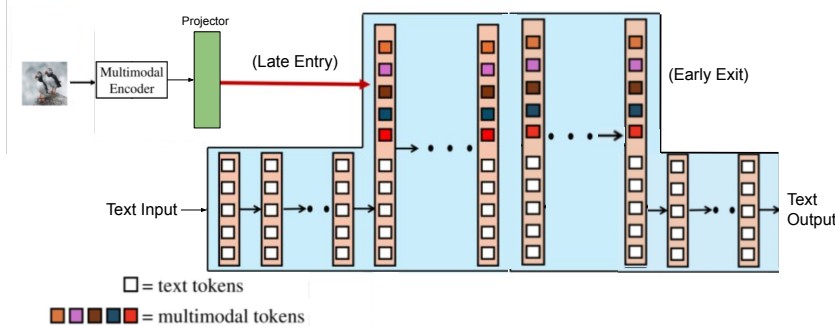

Figure 2: Early Exit and Late Entry. Specifically, the visual tokens are not passed into the first few layers but instead, directly inserted along with the prompt to the chosen layer for insertion or the vision tokens are removed from the forward pass after a certain layer.

**statement/proof of the sufficiency of redundancy and minimal vision-to-text attention for late-entry.]**

**Theorem 6** (Informal). *Given an insertion layer L, if the vision activations $V_l$, $l < L$, undergo minimal change and the effect of vision tokens on text representations is minimal then (assuming Lipschitz regularity conditions), the output with late entry of vision will be close to the true original output.*

*Proof.* We can bound the maximum transition of the vision representations across the $L$ layers via repeated applications of the triangle inequality. We can then bound the maximum transition between the correct text representation at layer $L$ and the representation from skipping vision using our assumption on minimal affect of vision tokens on text representations. Finally, using some Lipschitz regularity assumptions we arrive at a bound on the maximum error from skipping. Refer to Appendix A.2 for more details. □

Note that redundancy is clearly a necessary condition as if the visual representations have evolved too much than simple insertion of the original representations at layer $L$ will not necessarily give good outputs. On the other hand, for early exit we can see that just minimal cross attention from vision to text is sufficient because the output modality is text. Thus, if the vision is not communicating much with the text, it can be skipped. This is quantified using the Visual Attention Ratio (VAR) Jiang et al. (2025).

To find redundancies of a modality in layers, we aim to experimentally identify cases of significant geometric and proximal redundancy. Mathematically, this means that for $0 < \epsilon, t \ll 1$,

$\mathbb{E}[\rho(X_\ell, X_{\ell-1})] < \epsilon$ and $\mathbb{P}[\rho(X_\ell, X_{\ell-1}) < t] > 1 - \epsilon$. From Theorems 1 and 3 we know that these conditions will imply functional and informational redundancy.

### 4.1 VALIDATION OF CONDITIONS FOR LAYER SKIPPING

#### 4.1.1 EXPERIMENTAL SETUP

In this experiment, we consider the hidden states of each token across each layer. We compute both the average cosine distance and the probability of a small average cosine distance between adjacent hidden states. We separate vision and textual tokens so that we can evaluate their differences. Formally, let $H_T$ and $H_V$ denote the sets of textual and visual hidden states for all layers. For token $i$ at layer $\ell$, let $h_{\ell,i}$ denote its hidden state. We define the average cosine distance between adjacent layers as

$$\mathcal{D}_\ell^{(T)} := \frac{1}{N_T} \sum_{i=1}^{N_T} \rho\big(h_{\ell,i}^T, h_{\ell-1,i}^T\big), \quad \mathcal{D}_\ell^{(V)} := \frac{1}{N_V} \sum_{i=1}^{N_V} \rho\big(h_{\ell,i}^V, h_{\ell-1,i}^V\big), \tag{1}$$

and define the probability of being close as

$$p_\ell(t; H_T) := \frac{1}{N_T} \sum_{i=1}^{N_T} \mathbb{1}\{\rho\big(h_{\ell,i}^T, h_{\ell-1,i}^T\big) < t\}, \quad p_\ell(t; H_V) := \frac{1}{N_V} \sum_{i=1}^{N_V} \mathbb{1}\{\rho\big(h_{\ell,i}^V, h_{\ell-1,i}^V\big) < t\}, \tag{2}$$

where $t$ is a user-specified threshold, $N_T = |H_T^{(\ell)}|$ and $N_V = |H_V^{(\ell)}|$ are the number of textual and visual tokens, respectively, $h_{\ell,i}^T, h_{\ell-1,i}^T \in H_T$, $h_{\ell,i}^V, h_{\ell-1,i}^V \in H_V$, and $\rho(\cdot, \cdot)$ denotes cosine distance. This is done for $\ell \in [1, N]$, where $N$ is the number of layers in the specific model.

#### 4.1.2 MODELS & DATASETS

Both experiments use LLaVA 1.5 (7B and 13B) (Liu et al., 2023) and LLaVA NeXT (1.6) (Liu et al., 2024a) as VLMs. Additional results using DeepSeek-VL (7B base) (Lu et al., 2024) and Qwen 2.5 VL (Qwen et al., 2025) are in Appendix C.

The experiments are performed using multiple choice and free response datasets, spanning a diverse set of vision-language tasks including general question answering (GQA (Hudson & Manning, 2019), VQA (Agrawal et al., 2015), Visual7W (Zhu et al., 2016)), text, OCR, and document-based (AI2D (Kembhavi et al., 2016), OCRBench (Liu et al., 2024b), TextVQA (Singh et al., 2019)), and multimodal reasoning (MMMU (Yue et al., 2024), RealWorldQA (xAI, 2024), MMStar (Chen et al., 2024b), MathVision (Wang et al., 2024)). For further details on dataset and evaluation protocols, please refer to the Appendix B.

#### 4.1.3 RESULTS

Figure 4 shows that the early layer vision tokens have very low cosine distances with each other ($\rho(X_\ell, X_{\ell-1}) < \epsilon$) and very high probability of being close to each other (0.05-proximal with probability $> 1 - \epsilon$). In the later layers of LLaVA 1.5 7B and 13B, both the early and textual tokens demonstrate proximal redundancy ($t = 0.05$, probability $\geq 1 - \epsilon$). In LLaVA 1.6, this trend is visible but not as extreme. Note that the 0.05 threshold is arbitrary; we seek to highlight that early vision tokens have low high probabilities of being close, which means that there are redundancies via Theorems 1—4. Additionally, for each model, the adjacent distances and probabilities for both vision and textual tokens across all tested datasets are similar. This indicates that the model itself is more important in determining which layers can be dropped relative to the dataset itself. Refer to Appendix C for additional results. **[Additional results using Centralized Kernel Alignment (CKA) (Kornblith et al., 2019) can also be found in Appendix C]**.

The results in this section indicate that these models exhibit clear redundancy that can be exploited for efficiency improvements. We now validate that the inter-modal attention is sufficiently low to allow for late entry. As stated in Section 4, minimal vision-to-text attention is sufficient for early exit.

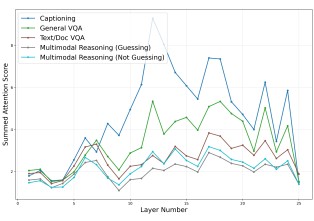 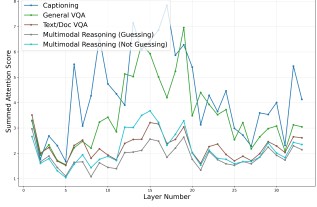 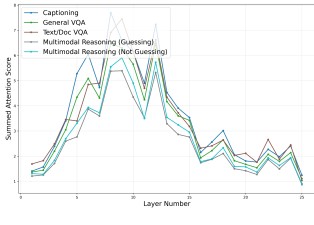

(a) VAR for LLaVA 1.5 7B      (b) VAR for LLaVA 1.5 13B      (c) VAR for LLaVA NeXT 7B

Figure 3: Visual Attention Ratio (VAR) (Jiang et al., 2025) with respect to the answer text token. In general, the vision-to-text attention is mainly in the middle layers, with the early and late layers having minimal vision-to-text attention.

#### 4.1.4 INTER-MODAL ATTENTION ANALYSIS

To determine layers viable for skipping, we analyze inter-modal attention in addition to redundancy. If inter-modality interaction is not minimal, then even if one modality has significant redundancy, its output is still necessary for processing in the other modality, so skipping is not viable. In Figure 3 we use the Visual Attention Ratio (VAR) from Jiang et al. (2025) to visualize the self-attention between vision and text. Specifically, they define VAR for each head $h$ at layer $\ell$ for the $k$-th text token $\mathbf{y}_k$ to be

$$\text{VAR}^{(\ell)}(\mathbf{y}_k) \triangleq \sum_{j=0}^{h} \sum_{i=1}^{n} \mathbf{A}_k^{(\ell,j)}(a_k, i) \tag{3}$$

where $\mathbf{A}_k^{(\ell,j)}(a_k, i)$ is the head-wise sum of the attention weights of the newly generated token $\mathbf{y}_k$ assigned to the image token $\mathbf{v}_i$. We, in particular, visualize just the VAR with respect to the answer token. From the plots in Figure 3 we see that in general across all datasets and models the early and late layers have minimal cross attention according to this metric in comparison to the middle layers. Previous work has described that this happens because the majority of visual information processing happens in these layers (Lin et al., 2025; Shang et al., 2025; Choraria et al., 2025; Jiang et al., 2025).

### 4.2 CONNECTING CONDITIONS TO PERFORMANCE DEGRADATION

In this section, **[we complete the case study by running layer skipping in two regions where we expect to see redundancy, thereby validating whether]** the conditions are directly related to model degradation. Based on our results in 4.1 and Theorems 1–4, we deduce that the early and late layers are often highly redundant with respect to visual information. **[Therefore, we layer skip the first $i$ vision tokens (late entry) and the last $j$ vision tokens (early stopping).]** We then compare the performance of these experiments to whether or not the conditions of redundancy are being met. In this way, we show that the proposed **[redundancy]** conditions can be **[used as a guide for which layers and tokens can be pruned out of VLMs by various techniques.]**

#### 4.2.1 RESULTS

The table below shows how the redundancy conditions relate to model degradation.

Based on Table 1, in Late entry layer skipping, the geometric and proximal redundancy is similar for the two Llava 1.5 models for multimodal reasoning and VQA tasks, however other models differ and the Captioning task. This implies that redundancy on Layer Skipping is model and task dependent.

For the early exit layer skipping, in Table 2, the VAR across modals differs, however minimally changes across tasks. Notably, however, VAR scores in the later layers on Captioning is much higher than the VQA/Mulimodal reasoning tasks.

| Key Takeaway #1 |
| --- |
| Early layer vision redundancy is high across models and datasets. |

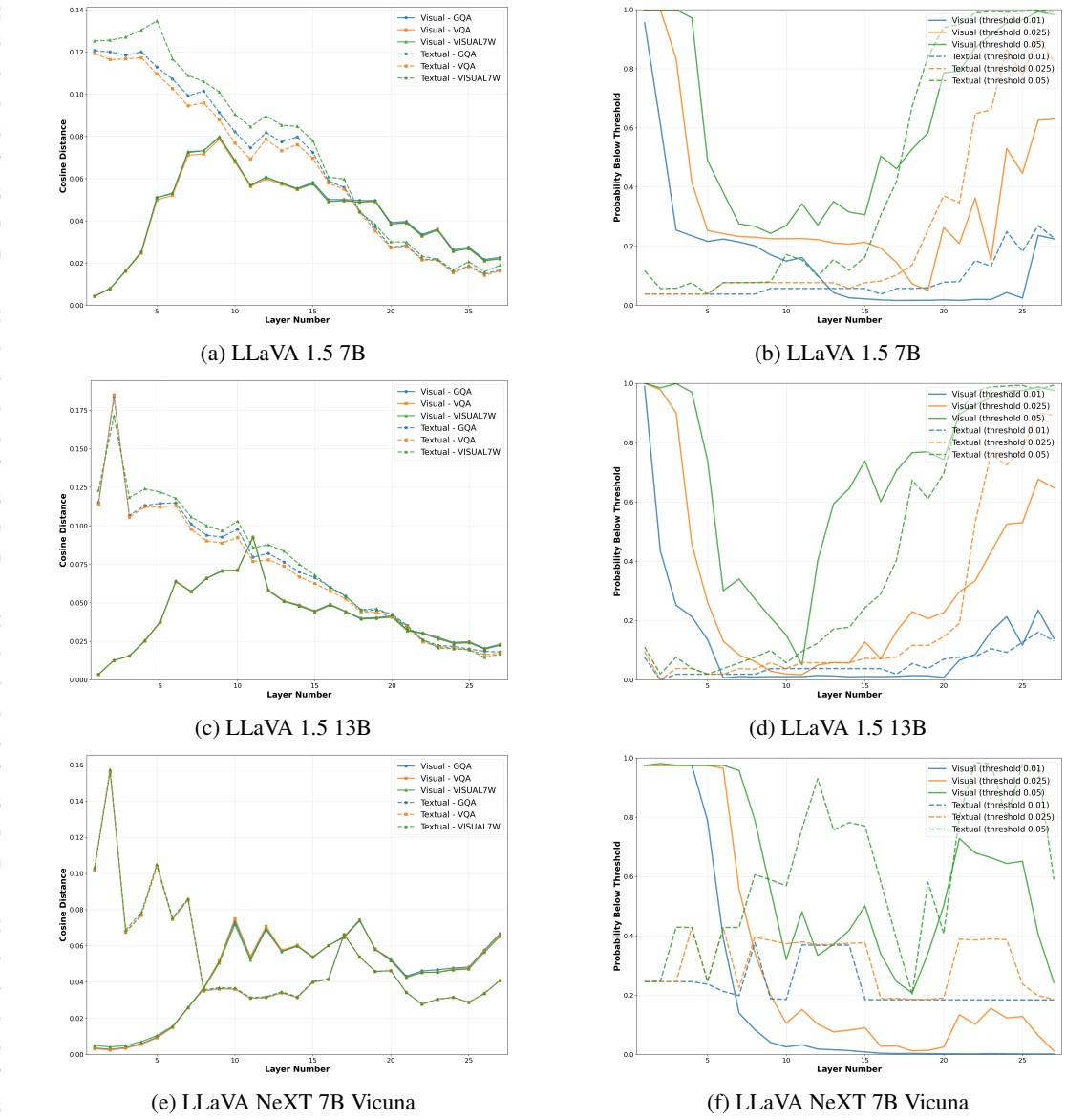

(a) LLaVA 1.5 7B         (b) LLaVA 1.5 7B

(c) LLaVA 1.5 13B         (d) LLaVA 1.5 13B

(e) LLaVA NeXT 7B Vicuna         (f) LLaVA NeXT 7B Vicuna

Figure 4: Empirical geometric and proximal redundancy experiments across layers for the LLaVA 1.5 7B/13B and LLaVA NeXT 7B. Across all the General VQA task (see Table 3) and models, the early layer vision tokens have low adjacent token cosine distances, and the textual and visual tokens have low adjacent token cosine distances in later layers.

**Key Takeaway #2**

Early layer textual redundancy is low in discriminative VQA tasks, but becomes low in layers after the Captioning tasks (See Figures 10 and 11 in the appendix).

**Key Takeaway #3**

VAR scores in the later layers is much higher on Captioning tasks than discriminative VQA.

**[In this section, we used the proposed redundancy framework to study layer skipping of early and late vision tokens. This case study represents an example of how the redundancy frame-**

| Task | Metric | LLaVA 1.5 7B | | | | LLaVA NeXT 7B Vicuna | | | | LLaVA 1.5 13B | | | | Qwen 2.5 VL | | | |
|---|---|---|---|---|---|---|---|---|---|---|---|---|---|---|---|---|---|
| | | 0 | 4 | 8 | 12 | 0 | 4 | 8 | 12 | 0 | 4 | 8 | 12 | 0 | 4 | 8 | 12 |
| General VQA | $\rho(V_\ell, V_{\ell-1})$ | – | 0.025 | 0.073 | 0.060 | – | 0.006 | 0.037 | 0.070 | – | 0.025 | 0.066 | 0.058 | – | 0.0323 | 0.050 | 0.0353 |
| | $\mathbb{P}[D_V < 0.05]$ | – | 0.972 | 0.267 | 0.271 | – | 0.974 | 0.791 | 0.334 | – | 0.965 | 0.273 | 0.403 | – | 0.9246 | 0.625 | 0.922 |
| | Accuracy | **0.564** | **0.553** | 0.370 | 0.261 | **0.770** | **0.721** | 0.630 | 0.514 | **0.782** | **0.779** | 0.747 | 0.5010 | **0.823** | 0.587 | 0.579 | 0.526 |
| Text/Doc VQA | $\rho(V_\ell, V_{\ell-1})$ | – | 0.020 | 0.061 | 0.058 | – | 0.007 | 0.039 | 0.037 | – | 0.019 | 0.059 | 0.059 | – | 0.506 | 0.0692 | 0.0444 |
| | $\mathbb{P}[D_V < 0.05]$ | – | 0.992 | 0.378 | 0.337 | – | 0.967 | 0.891 | 0.410 | – | 0.995 | 0.362 | 0.411 | – | 0.6515 | 0.337 | 0.70 |
| | Accuracy | **0.5790** | **0.5640** | 0.5130 | 0.4920 | **0.703** | 0.5400 | 0.4420 | 0.3320 | **0.6920** | **0.6780** | 0.6260 | 0.6000 | **0.804** | 0.710 | 0.549 | 0.513 |
| Multimodal Reasoning (Guessing) | $\rho(V_\ell, V_{\ell-1})$ | – | 0.019 | 0.056 | 0.053 | – | 0.011 | 0.033 | 0.061 | – | 0.019 | 0.062 | 0.060 | – | 0.057 | 0.085 | 0.047 |
| | $\mathbb{P}[D_V < 0.05]$ | – | 0.987 | 0.465 | 0.427 | – | 0.939 | 0.871 | 0.413 | – | 0.995 | 0.361 | 0.402 | – | 0.470 | 0.136 | 0.596 |
| | Accuracy | **0.255** | **0.250** | 0.242 | 0.228 | **0.231** | 0.143 | 0.105 | 0.088 | **0.261** | **0.256** | **0.253** | 0.195 | **0.350** | 0.224 | 0.244 | 0.180 |
| Multimodal Reasoning (Not Guessing) | $\rho(V_\ell, V_{\ell-1})$ | – | 0.025 | 0.072 | 0.059 | – | 0.008 | 0.036 | 0.067 | – | 0.024 | 0.066 | 0.058 | – | 0.044 | 0.071 | 0.045 |
| | $\mathbb{P}[D_V < 0.05]$ | – | 0.965 | 0.333 | 0.322 | – | 0.961 | 0.821 | 0.346 | – | 0.978 | 0.314 | 0.441 | – | 0.731 | 0.335 | 0.692 |
| | Accuracy | **0.325** | **0.321** | 0.302 | 0.233 | **0.384** | 0.248 | 0.258 | 0.155 | **0.341** | **0.343** | 0.316 | 0.245 | **0.637** | 0.412 | 0.401 | 0.325 |
| | | 0 | 8 | 12 | 16 | 0 | 8 | 12 | 16 | 0 | 8 | 12 | 16 | 0 | 8 | 12 | 16 |
| Captioning | $\rho(V_\ell, V_{\ell-1})$ | – | 0.027 | 0.0729 | 0.0603 | – | 0.006 | 0.0372 | 0.0710 | – | 0.027 | 0.0669 | 0.0589 | – | 0.035 | 0.0519 | 0.0369 |
| | $\mathbb{P}[D_V < 0.05]$ | – | 0.951 | 0.267 | 0.270 | – | 0.976 | 0.782 | 0.314 | – | 0.948 | 0.265 | 0.386 | – | 0.906 | 0.587 | 0.897 |
| | ClipSCORE | **0.308** | 0.154 | 0.227 | 0.214 | **0.312** | 0.153 | 0.154 | 0.141 | **0.3056** | 0.152 | 0.151 | 0.147 | **0.326** | 0.315 | 0.308 | 0.273 |
| | BLEU | **0.217** | **0.270** | **0.261** | 0.221 | 0.168 | **0.242** | 0.158 | **0.181** | 0.199 | 0.182 | **0.222** | **0.214** | **0.141** | 0.112 | 0.121 | 0.067 |
| | CIDEr | 0.819 | **0.902** | **0.872** | 0.720 | 0.583 | **0.766** | 0.521 | 0.596 | 0.526 | 0.464 | **0.651** | **0.678** | **0.395** | **0.374** | 0.303 | 0.155 |
| | SPICE | **0.213** | **0.215** | **0.197** | 0.163 | **0.171** | **0.179** | 0.161 | 0.147 | **0.200** | 0.186 | **0.207** | 0.189 | **0.166** | **0.1402** | **0.146** | 0.092 |

Table 1: Summary of results for layer skipping on vision tokens. In this table, we only focus on the first 16 layers of each model. After this point, skipping causes each model to degrade even further. **[We split up the Multimodal Reasoning Task into datasets in which the models have accuracy close to guessing in the dataset. This was done to show how redundancy is less useful when a model has very poor performance, such that there is no model degradation.]** In this table, accuracy in bold indicates closeness to the baseline performance.

| Task | Metric | LLaVA 1.5 7B | | | LLaVA NeXT 7B Vicuna | | | LLaVA 1.5 13B | | | | | Qwen 2.5 VL 7B | | |
|---|---|---|---|---|---|---|---|---|---|---|---|---|---|---|---|
| | | 20 | 24 | 28 | 20 | 24 | 28 | 20 | 24 | 28 | 32 | 36 | 20 | 24 | 28 |
| General VQA | VAR | 4.33 | 4.11 | 2.44 | 2.53 | 1.98 | 2.19 | 4.26 | 3.27 | 2.64 | 2.58 | 4.01 | 8.43 | 6.65 | 3.45 |
| | Accuracy | **0.582** | **0.584** | **0.581** | **0.647** | **0.647** | **0.647** | **0.787** | **0.787** | 0.780 | **0.787** | | **0.853** | **0.853** | **0.874** |
| Text/Doc VQA | VAR | 5.95 | 5.63 | 2.93 | 2.73 | 2.44 | 2.39 | 5.38 | 4.11 | 3.40 | 3.42 | 5.05 | 6.44 | 5.81 | 3.42 |
| | Accuracy | **0.592** | **0.594** | **0.595** | **0.684** | **0.701** | **0.701** | **0.691** | **0.710** | **0.708** | **0.708** | | **0.883** | **0.882** | **0.912** |
| Multimodal Reasoning (Guessing) | VAR | 2.67 | 2.34 | 1.87 | 2.11 | 1.86 | 1.78 | 2.21 | 2.07 | 1.67 | 1.92 | 2.82 | 9.01 | 6.08 | 3.23 |
| | Accuracy | **0.248** | **0.248** | **0.248** | **0.234** | **0.237** | **0.238** | 0.245 | **0.263** | **0.262** | **0.264** | 0.263 | **0.243** | 0.221 | **0.254** |
| Multimodal Reasoning (Not Guessing) | VAR | 3.00 | 2.60 | 1.97 | 2.33 | 1.93 | 1.92 | 2.75 | 2.13 | 1.67 | 2.01 | 3.14 | 1.49 | 1.05 | 1.01 |
| | Accuracy | **0.322** | **0.325** | **0.326** | 0.343 | 0.341 | 0.342 | 0.313 | **0.335** | **0.335** | **0.335** | 0.333 | 0.533 | 0.530 | **0.593** |
| | | 20 | 24 | 28 | 20 | 24 | 28 | 16 | 20 | 24 | 28 | 32 | 20 | 24 | 28 |
| Captioning | VAR | 7.35 | 6.24 | 3.21 | 3.01 | 2.27 | 2.62 | 5.87 | 4.29 | 2.72 | 4.00 | 5.74 | 8.49 | 3.56 | 1.32 |
| | ClipSCORE | **0.3061** | **0.3072** | **0.308** | **0.3125** | **0.3127** | **0.3184** | 0.294 | **0.3014** | **0.3045** | **0.3038** | **0.3039** | 0.294 | **0.318** | **0.322** |
| | BLEU | **0.203** | **0.219** | **0.216** | **0.165** | **0.168** | **0.168** | 0.143 | **0.161** | **0.184** | **0.194** | 0.143 | 0.069 | **0.107** | **0.121** |
| | CIDEr | 0.573 | **0.620** | **0.625** | **0.561** | **0.560** | **0.566** | 0.377 | 0.433 | **0.499** | **0.509** | **0.511** | 0.185 | **0.320** | **0.358** |
| | SPICE | **0.206** | **0.213** | **0.213** | **0.163** | **0.164** | **0.165** | 0.160 | 0.176 | **0.193** | **0.195** | **0.197** | 0.094 | **0.138** | **0.152** |

Table 2: Summary of results for early stopping on vision tokens. In this table, we only focus on the layers starting on the 20th layer of each model. Before this point, early skipping causes significant performance degradation across models due to being to visual information still being processed, which is supported in Figure 3. We include layer 20 to see how are framework is applicable when VAR is moderate). **[We split up the Multimodal Reasoning Task into datasets in which the models have accuracy close to guessing in the dataset. This was done to show how redundancy is less useful when a model has very poor performance, such that there is no model degradation.]** In this table, accuracy in bold indicates closeness to the baseline performance.

**work can be used to better understand information processing and inspire token and layer reduction methods.]**

## 5 DISCUSSION

**[Our redundancy framework is incredibly general as it largely relies on probability and information theory. Thus, in theory, it can provide a unified foundation for analyzing a wide range of pruning techniques. To demonstrate its flexibility, consider a few recent examples. First, our results offer some theoretical justification for claims of layer-level redundancy, such as those made by Shukor & Cord (2024). Secondly, Skip-Vision (Zeng et al., 2025) employs a token-merging strategy based on cosine similarity in which tokens are ranked by similarity and merged to reduce redundancy. Additionally they propose the *skip-FNN* module, which skips certain redundant feed-forward computations. Our framework applies directly to both strategies. In fact, their theoretical justification for skipping is similar to our functional redundancy theorems. FlexiDepth (Luo et al., 2025a) can also be interpreted as effectively learning $\mathbb{E}[Z \mid X_\ell]$ at each layer $\ell$. Finally, ST[3] (Zhuang et al., 2024) identifies "lazy" layers, that is, layers which minimally contribute beyond their previous layer. These "lazy" layers are determined using cosine similarity, a scenario that our framework can also analyze directly.]**

## 6 CONCLUSION & FUTURE WORK

In this work, we propose a theoretical framework to study redundancies in VLMs. We then demonstrate that empirically verifiable notions of redundancy, namely average cosine distance and small cosine distance, with high probability imply more informative notions of redundancy. By combining these results with inter-modal attention analysis, we identify redundant layers in the model with respect to multimodal processing and experimentally show that these redundant layers appear in the early vision tokens, late vision tokens, and late textual tokens. We then validate these findings by skipping these layers and finding minimal performance degradation across different tasks. Conversely, we also show that skipping non-redundant layers severely degrades model performance.

Future work includes further validation our results by expanding the scope of our experiments on more datasets and models. Additionally, it would be interesting to study the factors underlying dramatic instances of VLM hallucination, based on recent observations and datasets in literature (Vo et al., 2025), through the lens of our framework. A longer term pursuit would perhaps be understanding why these redundancies exist and whether they arise as a drawback of vision-language pretraining strategies, or as an intentional mechanism for multimodal processing.

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

APPENDIX

# A PROOFS

## A.1 PROOFS OF LEMMAS

**Lemma 1.** *Let $(X, d)$ be a metric space and $x, y, z \in X$. Then $d(x,y)^2 \leq 2d(x,z)^2 + 2d(z,y)^2$.*

*Proof.* Follows from the triangle inequality and the AM-GM inequality. $\square$

**Lemma 2.** *Suppose $X, Y$ are random vectors with unit norm (with probability 1). Let $\rho(x, y) = 1 - \frac{\langle x, y \rangle}{\|x\| \|y\|}$ be the cosine distance between $x$ and $y$. Then $\mathbb{E}[\rho(X, Y)] < \frac{\epsilon}{2}$ implies $\mathbb{E}[\|X - Y\|_2^2] < \epsilon$.*

*Proof.* Observe that

$$\mathbb{E}[\|X - Y\|_2^2] = \mathbb{E}[\|X\|^2 + \|Y\|^2 - 2\langle X, Y \rangle] = 2\mathbb{E}[1 - \langle X, Y \rangle] \tag{4}$$

$$= 2\mathbb{E}[\rho(X, Y)] \tag{5}$$

$$< \epsilon. \tag{6}$$

$\square$

**Lemma 3.** *Let $(X, \langle \cdot, \cdot \rangle)$ be a real inner product space and $a, b, c \in X$. Then $\|a + b + c\|^2 \leq 3(\|a\|^2 + \|b\|^2 + \|c\|^2)$.*

*Proof.* Expanding $\|a + b + c\|^2$ we get $\|a\|^2 + \|b\|^2 + \|c\|^2 + 2\langle a, b \rangle + 2\langle b, c \rangle + 2\langle a, c \rangle$. Thus we want to show that $0 \leq 2\|a\|^2 + 2\|b\|^2 + 2\|c\|^2 - 2\langle a, b \rangle - 2\langle b, c \rangle - 2\langle a, c \rangle$. This is equivalent to showing $0 \leq \|a - b\|^2 + \|b - c\|^2 + \|a - c\|^2$ and since $\| \cdot \|^2$ is non-negative, we are done. $\square$

**Lemma 4.** *Suppose $Y{-}X{-}Z$ is a Markov chain. Then $\mathbb{E}_{(X,Y)}[D(p_{Z|X}\|p_{Z|Y})] = I(Z; X|Y)$.*

*Proof.* Observe that

$$\mathbb{E}[D(p_{Z|X}\|p_{Z|Y})] = \mathbb{E}\left[\int p_{Z|X}(z|X) \log \frac{p_{Z|X}(z|X)}{p_{Z|Y}(z|Y)} dz\right] \tag{7}$$

$$= \mathbb{E}_{(X,Y), Z \sim p(\cdot|X)}\left[\log \frac{p(Z|X)}{p(Z|Y)}\right] \tag{8}$$

$$= \mathbb{E}_{(X,Y), Z \sim p(\cdot|X,Y)}\left[\log \frac{p(Z|X,Y)}{p(Z|Y)}\right] \tag{9}$$

$$= \mathbb{E}_{X,Y,Z}\left[\log \frac{p(Z, X|Y)}{p(X|Y)p(Z|Y)}\right] \tag{10}$$

$$= I(Z; X|Y) \tag{11}$$

where equation 9 follows from the Markov property and equation 11 is the definition of conditional mutual information. $\square$

## A.2 PROOFS OF PROPOSITIONS AND THEOREMS

**Proposition 1.** *Let $\rho : \mathcal{X} \times \mathcal{X} \to \mathbb{R}$ be a symmetric function with $0 \leq \rho \leq 1$. Then*

$$\mathbb{P}[\rho(X, Y) > t] < \frac{\epsilon - t}{1 - t} \;\; \text{implies} \;\; \mathbb{E}[\rho(X, Y)] < \epsilon$$

*Proof.* Define the random variable $D := \rho(X, Y)$. By the tail integration formula,

$$\mathbb{E}[D] = \int_0^1 \mathbb{P}[D > s]ds \tag{12}$$

$$= \int_0^t \mathbb{P}[D > s]ds + \int_t^1 \mathbb{P}[D > s]ds \tag{13}$$

$$\leq t \cdot 1 + (1 - t)\mathbb{P}[D > t] \tag{14}$$

$$< t + (1 - t)(\frac{\epsilon - t}{1 - t}) \tag{15}$$

$$= \epsilon. \tag{16}$$

$\square$

**Theorem 1.** *Let $X_\ell, X_{\ell-1}$ be unit-norm random variables and $Z$ be another random variable (e.g. hidden representations of layers $\ell, \ell - 1$ and the task ground truth respectively). Let $\rho(x, y) = 1 - \frac{\langle x, y \rangle}{\|x\|\|y\|}$ .Assume $\mathbb{E}[\rho(X, Y)] < \frac{\epsilon}{2}$ and that*

$$h(x, y) = E[Z|X_\ell = x, X_{\ell-1} = y].$$

*is $\alpha$-Lipschitz in the first argument and $\beta$-Lipschitz in the second. Then $E[\|E[Z|X_\ell] - E[Z|X_{\ell-1}]\|_2^2] < 2(\alpha^2 + \beta^2)\epsilon$.*

*Proof.* Observe that

$$\mathbb{E}[Z|X_\ell] - E[Z|X_{\ell-1}] = \mathbb{E}[\mathbb{E}[Z|X_\ell, X_{\ell-1}]|X_\ell] - \mathbb{E}[\mathbb{E}[Z|X_\ell, X_{\ell-1}]|X_{\ell-1}] \tag{17}$$

$$= \mathbb{E}[h|X_\ell] - \mathbb{E}[h|X_{\ell-1}] \tag{18}$$

By Lemma 2 we have that $\mathbb{E}[\|X_\ell - X_{\ell-1}\|_2^2] < \epsilon$ since $\|X_\ell\|$ and $\|X_{\ell-1}\|$ are unit-norm. By Lemma 1 we have

$$(\mathbb{E}[h|X_\ell] - \mathbb{E}[h|X_{\ell-1}])^2 \leq 2(\mathbb{E}[h|X_\ell] - h)^2 + 2(\mathbb{E}[h|X_{\ell-1}] - h)^2$$

Furthermore, since expectation is order-preserving, we have

$$\mathbb{E}[(\mathbb{E}[h|X_\ell] - \mathbb{E}[h|X_{\ell-1}])^2] \leq 2\mathbb{E}[(\mathbb{E}[h|X_\ell] - h)^2] + 2\mathbb{E}[(\mathbb{E}[h|X_{\ell-1}] - h)^2] \tag{19}$$

$$= 2\mathbb{E}[\text{Var}(h|X_\ell)] + 2\mathbb{E}[\text{Var}(h|X_{\ell-1})] \tag{20}$$

By definition,

$$\mathbb{E}[\text{Var}(h(X_\ell, X_{\ell-1})|X_\ell = x)] = \mathbb{E}[\mathbb{E}[(h(X_\ell, X_{\ell-1}) - \mathbb{E}[h(X_\ell, X_{\ell-1})|X_\ell = x])^2|X_\ell = x]] \tag{21}$$

$$\leq \mathbb{E}[\mathbb{E}[(h(X_\ell, X_{\ell-1}) - h(X_\ell, X_{\ell-1} = \mathbb{E}[X_{\ell-1}|X_\ell = x]))^2|X_\ell = x]] \tag{22}$$

$$\leq \beta^2\mathbb{E}[\mathbb{E}[\|X_{\ell-1} - \mathbb{E}[X_{\ell-1}|X_\ell = x]\|_2^2|X_\ell = x]] \tag{23}$$

$$= \beta^2\mathbb{E}[\|X_{\ell-1} - \mathbb{E}[X_{\ell-1}|X_\ell = x]\|_2^2] \tag{24}$$

$$\leq \beta^2\mathbb{E}[\|X_{\ell-1} - X_\ell\|_2^2] \tag{25}$$

$$< \beta^2\epsilon \tag{26}$$

where equation 22 holds by the optimality of the minimum mean squared error (MMSE) estimator, equation 23 holds by the Lipschitz assumption, equation 24 holds by the tower property (Law of Total Expectation), and equation 25 holds by the optimality of the MMSE estimator once again. By symmetry, the same holds $X_{\ell-1}$ (i.e. $\mathbb{E}[\text{Var}(h|X_{\ell-1} = y)] < \alpha^2\epsilon$) so $\mathbb{E}[(\mathbb{E}[Z|X_\ell] - \mathbb{E}[Z|X_{\ell-1}])^2] \leq 2(\alpha^2 + \beta^2)\epsilon$ $\square$

**Theorem 2.** *Let $X_\ell, X_{\ell-1}$ be unit-norm random variables and $Z$ be another random variable (e.g. layer activations of layers $\ell, \ell - 1$ and a task variable respectively). Let $\rho(x, y) = 1 - \frac{\langle x, y \rangle}{\|x\|\|y\|}$ . Assume $\mathbb{E}[\rho(X, Y)] < \frac{\epsilon}{2}$ and that*

$$h(x, y) = E[Z|X_\ell = x, X_{\ell-1} = y]$$

*is $\alpha$-Lipschitz in the first argument and $\beta$-Lipschitz in the second. Let $\hat{f}_\ell$ is a finite-sample estimate of $f_\ell^*(x) = \mathbb{E}[Z|X_\ell = x]$ and $\hat{f}_{\ell-1}(x)$ is a finite-sample estimate of $f_{\ell-1}^*(x) = \mathbb{E}[Z|X_{\ell-1} = x]$. Further let, $\eta_\ell = \mathbb{E}[\|\hat{f}_\ell(X_\ell) - f_\ell^*(X_\ell)\|^2]$ and $\eta_{\ell-1} = \mathbb{E}[\|\hat{f}_{\ell-1}(X_{\ell-1}) - f_{\ell-1}^*(X_{\ell-1})\|^2]$. We then have*

$$\mathbb{E}[\|\hat{f}_\ell(X_\ell) - \hat{f}_{\ell-1}(X_{\ell-1})\|^2] < 3\eta_\ell + 3\eta_{\ell-1} + 6(\alpha^2 + \beta^2)\epsilon$$

.

*Proof.* By Lemma 2 we have that $\mathbb{E}[\|X_\ell - X_{\ell-1}\|_2^2] < \epsilon$.

Observe that

$$\hat{f}_\ell(X_\ell) - \hat{f}_{\ell-1}(X_{\ell-1}) = (-f_\ell^*(X_\ell) + \hat{f}_\ell(X_\ell)) + (f_{\ell-1}^*(X_{\ell-1}) - \hat{f}_{\ell-1}(X_{\ell-1})) + (f_\ell^*(X_\ell) - f_{\ell-1}^*(X_{\ell-1})).$$

Thus by Lemma 3 we have:

$$\|\hat{f}_\ell(X_\ell) - \hat{f}_{\ell-1}(X_{\ell-1})\|^2 \le 3(\|f_\ell^*(X_\ell) - \hat{f}_\ell(X_\ell)\|^2 + \|f_{\ell-1}^*(X_{\ell-1}) - \hat{f}_{\ell-1}(X_{\ell-1})\|^2 + \|f_{\ell-1}^*(X_{\ell-1}) - f_\ell^*(X_\ell)\|^2)$$

Finally, by taking expectations we get

$$\mathbb{E}[\|\hat{f}(X_\ell) - \hat{f}(X_{\ell-1})\|^2] \le 3(\eta_\ell + \eta_{\ell-1} + 2(L_1^2 + L_2^2)\epsilon) = 3\eta_\ell + 3\eta_{\ell-1} + 6(\alpha^2 + \beta^2)\epsilon.$$

$\square$

**Theorem 3** (Continuous Fano's Inequality; (Duchi & Wainwright, 2013))**.** *Let $X_\ell, X_{\ell-1}$ be unit-norm vectors over the support $\mathcal{X}$. Define*

$$\overline{\mathbb{B}}_\rho(t) = \{x' \in \mathbb{R}^d | \rho(x, x') \le t\}.$$

*Let $\mu$ be the Lebesgue measure. Assume $\mu(\partial\mathcal{X})$ and $\sup_{x \in \mathcal{X}} \mu(\partial(\mathbb{B}_\rho(t) \cap \mathcal{X}))$ are finite where the Lebesgue measure is taken over their respective dimensions. Let $P_t = \mathbb{P}[\rho(X_\ell, X_{\ell-1}) \ge t]$. Then if $X_\ell$ is uniform over $\mathcal{X}$,*

$$I(X_\ell, X_{\ell-1}) \ge (1 - P_t) \log\left(\frac{\mu(\mathcal{X})}{\sup_{x \in \mathcal{X}} \mu(\mathbb{B}_\rho(t) \cap \mathcal{X})}\right) - \log 2.$$

*Proof.* Observe that $X_\ell$—$X_{\ell-1}$—$X_{\ell-1}$ is trivially a Markov chain. The result follows from applying results from Duchi & Wainwright (2013). $\square$

**Theorem 4** ((Braun & Pokutta, 2015))**.** *Let $X_\ell, X_{\ell-1}$ be unit-norm random variables with shared support $\mathcal{X}$. Let $\rho : \mathcal{X} \times \mathcal{X} \to \mathbb{R}$ be a symmetric function (e.g. a metric). Let $\overline{B}(t, x) := \{x' \in \mathcal{X} | \rho(x, x') \le t\}$, $P_t = \mathbb{P}[\rho(X_{\ell-1}, X_\ell) > t]$. Let*

$$p_{min} := \inf_{x \in \mathcal{X}} \mathbb{P}[(X_{\ell-1}, x) \in \overline{B}(t, x)] \quad and \quad p_{max} := \sup_{x \in \mathcal{X}} \mathbb{P}[(X_{\ell-1}, x) \in \overline{B}(t, x)]$$

*with $0 \le p_{min} < 1$ and $0 < p_{max} \le 1$ and $p_{min} + p_{max} < 1$. Then,*

$$I(X_{\ell-1}; X_\ell) \ge (1 - P_t) \log \frac{1}{p_{max}} - P_t \log(1 - p_{min}) - H_2(P_t)$$

*Proof.* Follows directly from Proposition 2.2 in Braun & Pokutta (2015) with $R = \{(x, x') \in \mathcal{X} \times \mathcal{X} : \rho(x, x') \le t\}$. $\square$

**Corollary 4** (Conditional Entropy Fano's Inequality; (Braun & Pokutta, 2015))**.** *With the same conditions and notations as Theorem 4 we have*

$$H(X_\ell | X_{\ell-1}) \le H(X_\ell) + \log p_{max} + H(P_t) + P_t \log \frac{1 - p_{min}}{p_{max}}$$

*Proof.* Follows from Corollary 2.3 in Braun & Pokutta (2015) with $R = \{(x, x') \in \mathcal{X} \times \mathcal{X} : \rho(x, x') \le t\}$. $\square$

**Theorem 5.** *Suppose there are random variables $Z, X_\ell, X_{\ell-1}$ with $Z \in \mathbb{R}^d$ and $X_\ell, X_{\ell-1}$ continuous unit-norm random variables. Further suppose $\|Z\|_2 \leq B$ almost surely and that $X_{\ell-1}$— $X_\ell$— $Z$ is a Markov chain. Then $\mathbb{E}[\|\mathbb{E}[Z|X_\ell] - \mathbb{E}[Z|X_{\ell-1}]\|_2^2] \leq 2B^2 I(Z; X_\ell|X_{\ell-1})$.*
*If, in addition, there exists finite $C$ such that $H(X_\ell|Z, X_{\ell-1}) \geq -C$ then $\mathbb{E}[\|\mathbb{E}[Z|X_\ell] - \mathbb{E}[Z|X_{\ell-1}]\|_2^2] \leq 2B^2(H(X_\ell|X_{\ell-1}) + C)$. In particulary, if $X_\ell$ is discrete then $C = 0$ and if $p_{X_\ell|Z, X_{\ell-1}}(x) \leq M \; \forall x$ then $C = \log M$.*

*Proof.* Fix an $a, b$ and consider the conditional probability distributions $p_{Z|X_\ell=a}$ and $p_{Z|X_{\ell-1}=b}$. We then have that

$$\mathbb{E}[Z|X_\ell = a] - \mathbb{E}[Z|X_{\ell-1} = b] = \int_{\mathbb{R}^d} z(p_{Z|X_\ell=a}(z) - p_{Z|X_{\ell-1}=b}(z))dz.$$

Thus,

$$\|\mathbb{E}[Z|X_\ell = a] - \mathbb{E}[Z|X_{\ell-1} = b]\|_2 = \left\| \int_{\mathbb{R}^d} z(p_{Z|X_\ell=a}(z) - p_{Z|X_{\ell-1}=b}(z))dz \right\|_2 \quad (27)$$

$$\leq \int_{\mathbb{R}^d} \|z\|_2 |p_{Z|X_\ell=a}(z) - p_{Z|X_{\ell-1}=b}(z)|dz \quad (28)$$

$$\leq B \int_{\mathbb{R}^d} |p_{Z|X_\ell=a}(z) - p_{Z|X_{\ell-1}=b}(z)|dz \quad (29)$$

$$= 2B\delta_{TV}(p_{Z|X_\ell=a}, p_{Z|X_{\ell-1}=b}) \quad (30)$$

where equation 28 holds by the triangle inequality. Thus we have,

$$\mathbb{E}[\|\mathbb{E}[Z|X_\ell] - \mathbb{E}[Z|X_{\ell-1}]\|_2^2] \leq 4B^2 \mathbb{E}[\delta_{TV}(p_{Z|X_\ell=a}, p_{Z|X_{\ell-1}=b})^2] \quad (31)$$

$$\leq 2B^2 \mathbb{E}[D(p_{Z|X_\ell}||p_{Z|X_{\ell-1}})]. \quad (32)$$

where equation 32 holds by Pinsker's inequality.

Now, by Lemma 4 we have $2B^2 \mathbb{E}[D(p_{Z|X_\ell}||p_{Z|X_{\ell-1}})] = 2B^2 I(Z; X_\ell|X_{\ell-1})$. Finally, we know that $I(Z; X_\ell|X_{\ell-1}) = H(X_\ell|X_{\ell-1}) - H(X_\ell|Z, X_{\ell-1}) \leq H(X_\ell|X_{\ell-1}) + C$. Thus,

$$\mathbb{E}[\|\mathbb{E}[Z|X_\ell] - \mathbb{E}[Z|X_{\ell-1}]\|_2^2] \leq 2B^2 H(X_\ell|X_{\ell-1}) + C.$$

$\square$

**Theorem 6.** *Let $f = f^n \circ f^{n-1} \circ \cdots \circ f^1$ be the $n$ layers in a VLM. Fix a layer $L \in \{1, 2, \ldots, n\}$. Let*

$$X_l = \begin{pmatrix} V_l \\ T_l \end{pmatrix} = \begin{pmatrix} f_{vis}^l(V_{l-1}, T_{l-1}) \\ f_{text}^l(V_{l-1}, T_{l-1}) \end{pmatrix} = (f^l \circ f^{l-1} \circ \cdots f^1)\left( \begin{pmatrix} V_{l-1} \\ T_{l-1} \end{pmatrix} \right)$$

*and $X_0 = X = \begin{pmatrix} V \\ T \end{pmatrix}$ be the input. Let $\phi = f^n \circ f^{n-1} \circ \cdots f^{L+1}$ be the "tail" of the VLM. Let $Y_{true} = f(X)$ and $Y_{skip} = \phi\left( \begin{pmatrix} V_1 \\ \tilde{T}_L \end{pmatrix} \right)$ where $\tilde{T}_l = f_{text}^l(0, T_{l-1})$ be the text prediction at layer $l$ without visual information. Assume $\phi$ is $\mu$-Lipschitz and $f_{text}^l$ is $\lambda$-Lipschitz in the second argument for all $l \leq L$. Further assume that*

$$\|V_1 - V_2\|, \|V_2 - V_3\|, \ldots, \|V_{L-1} - V_L\| \leq \epsilon$$

*and*

$$\|f_{text}^l(0, T) - f_{text}^l(V_{l-1}, T)\| \leq \delta.$$

*Then*

$$\|Y_{true} - Y_{skip}\| \leq \mu((L-1)\epsilon + \delta(\frac{\lambda^L - 1}{\lambda - 1}))$$

*Proof.* By repeated application of the triangle inequality, we have $\|V_1 - V_L\| \leq (L-1)\epsilon$. Define

$$E_l := \|T_l - \tilde{T}_l\| = \|f_{text}^l(V_{l-1}, T_{l-1}) - f_{text}^l(0, \tilde{T}_{l-1})\| \quad (33)$$

$$\leq \|f_{text}^l(V_{l-1}, T_{l-1}) - f_{text}^l(0, T_{l-1})\| + \|f_{text}^l(0, T_{l-1}) - f_{text}^l(0, \tilde{T}_{l-1})\| \quad (34)$$

$$\leq \delta + \lambda\|T_{l-1} - \tilde{T}_{l-1}\| \quad (35)$$

$$= \delta + \lambda E_{l-1} \quad (36)$$

We thus have a linear recurrence relation which gives rise to a closed form solution of

$$E_L \leq \delta \left( \frac{\lambda^L - 1}{\lambda - 1} \right)$$

Thus,

$$\|Y_{true} - Y_{skip}\| = \|\phi(V_L, T_L) - \phi(V_1, \tilde{T}_L)\| \tag{37}$$

$$\leq \mu \| \begin{pmatrix} V_L \\ T_L \end{pmatrix} - \begin{pmatrix} V_1 \\ \tilde{T}_L \end{pmatrix} \| \tag{38}$$

$$\leq \mu \sqrt{\|V_L - V_1\|^2 + \|T_L - \tilde{T}_L\|^2} \tag{39}$$

$$\leq \mu \sqrt{ \left( ((L-1)\epsilon)^2 + \left( \delta \left( \frac{\lambda^L - 1}{\lambda - 1} \right) \right)^2 \right) } \tag{40}$$

$\square$

# B  DATASETS

In this work, we experiment on General Visual Question Answering (VQA), Text/Doc VQA, Multimodal Reasoning, and Math Reasoning. See the table below for a dataset breakdown.

| Task | Datasets |
|------|----------|
| General VQA | GQA, VQA, Visual7W |
| Text/Doc VQA | AI2D, OCRBench, TextVQA |
| Multimodal Reasoning | MMMU, RealWorldQA, MMStar, MathVision |
| Image Captioning | Coco, Flickr30k |

Table 3: Dataset Organization by task

## B.1  EVALUATION METHOD OF EACH DATASET

We split our datasets into two groups depending on whether they contain MCQ questions that can be answered in one token or not. The MCQ datasets include Visual7W, AI2D, MMMU, MMStar, and a subset of MathVision. We used an LLM-as-a-judge approach to evaluate the other datasets. These include: VQA, GQA, TextVQA, OCRBench, RealWorldQA, and a subset of MathVision.

For the MCQ datasets, we ran a forward pass to generate exactly the predicted letter (A, B, C, D). We then directly compared the predicted letter to the correct letter. Some of the datasets included Yes/No questions, and these were evaluated the same way.

For the LLM-as-a-judge datasets, we generated 256 tokens. We then used GPT-5 (OpenAI, 2025) to evaluate if the predicted answer was correct given the question and correct answer. This approach was beneficial to avoid association reasoning problems that smaller models (13B or less parameters) may have answering complex questions.

# C  FURTHER RESULTS FROM SECTION 4

We include experiments on Text/Doc VQA, Multimodal Reasoning, and Captioning datasets on all models to validate our results are consistent across task types.

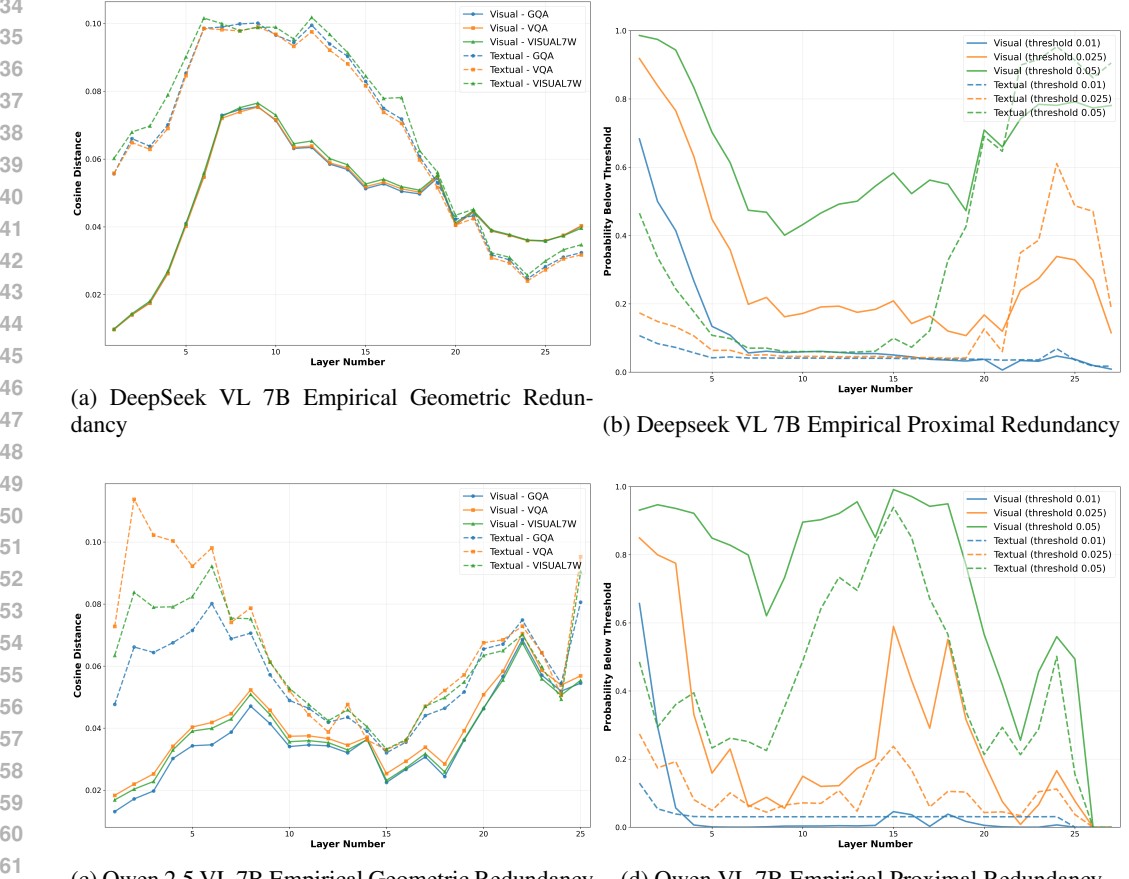

(a) DeepSeek VL 7B Empirical Geometric Redundancy

(b) Deepseek VL 7B Empirical Proximal Redundancy

(c) Qwen 2.5 VL 7B Empirical Geometric Redundancy

(d) Qwen VL 7B Empirical Proximal Redundancy

Figure 5: Empirical Geometric and Proximal Redundancy versus layer for the Qwen 2.5 VL and Deepseek VL 7B VLMs. Across all datasets in the General VQA task (see Table 3) and models, the early layer vision tokens have low adjacent token cosine distances, and the textual and visual tokens have low adjacent token cosine distances in later layers.

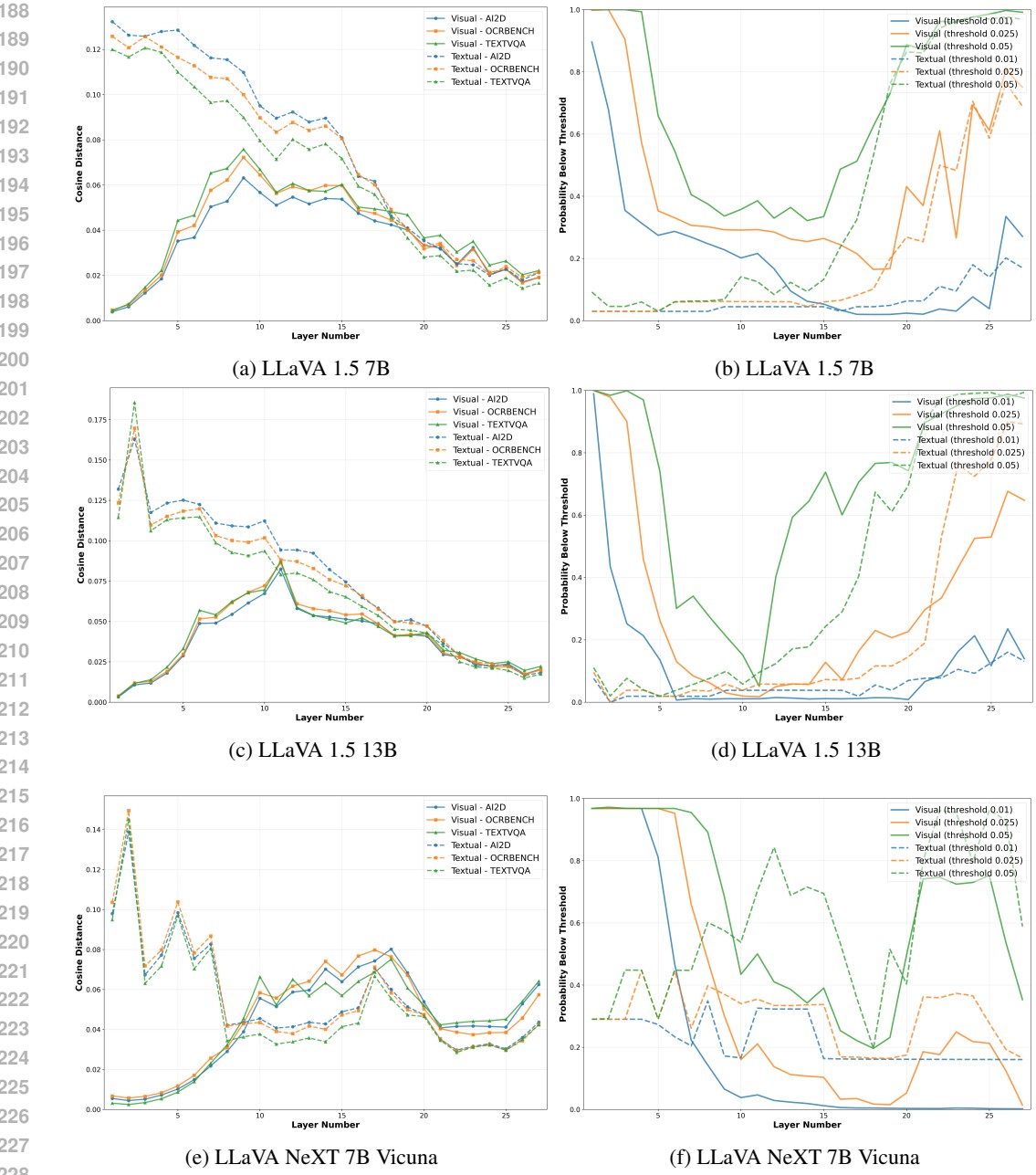

Figure 6: Empirical Geometric and Proximal Redundancy versus layer for the LlaVA models. Across all datasets in the Text/Doc VQA task (see Table 3) and models, the early and late layer vision tokens have low adjacent token cosine distances.

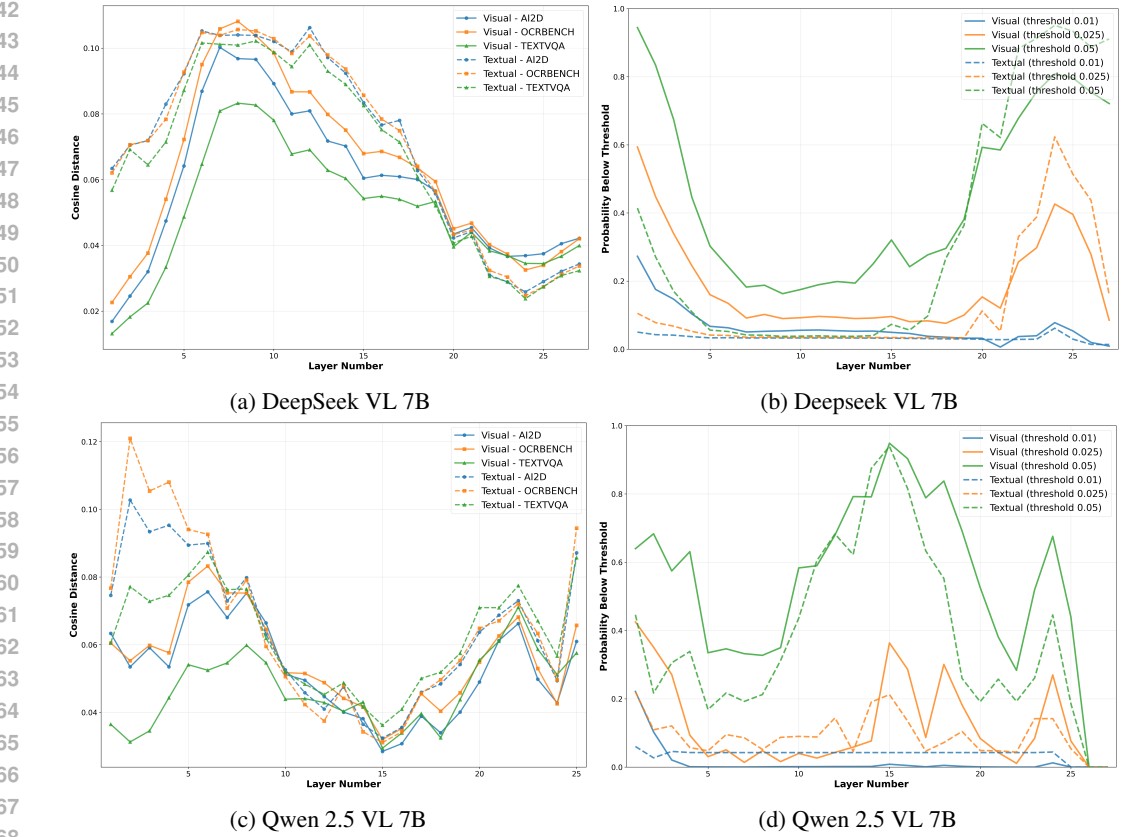

(a) DeepSeek VL 7B

(b) Deepseek VL 7B

(c) Qwen 2.5 VL 7B

(d) Qwen 2.5 VL 7B

Figure 7: Empirical Geometric and Proximal Redundancy versus layer for the Qwen 2.5 VL and Deepseek VL 7B VLMs. Across all datasets in the Text/Doc VQA task (see Table 3) and models, the early and late layer vision tokens have low adjacent token cosine distances.

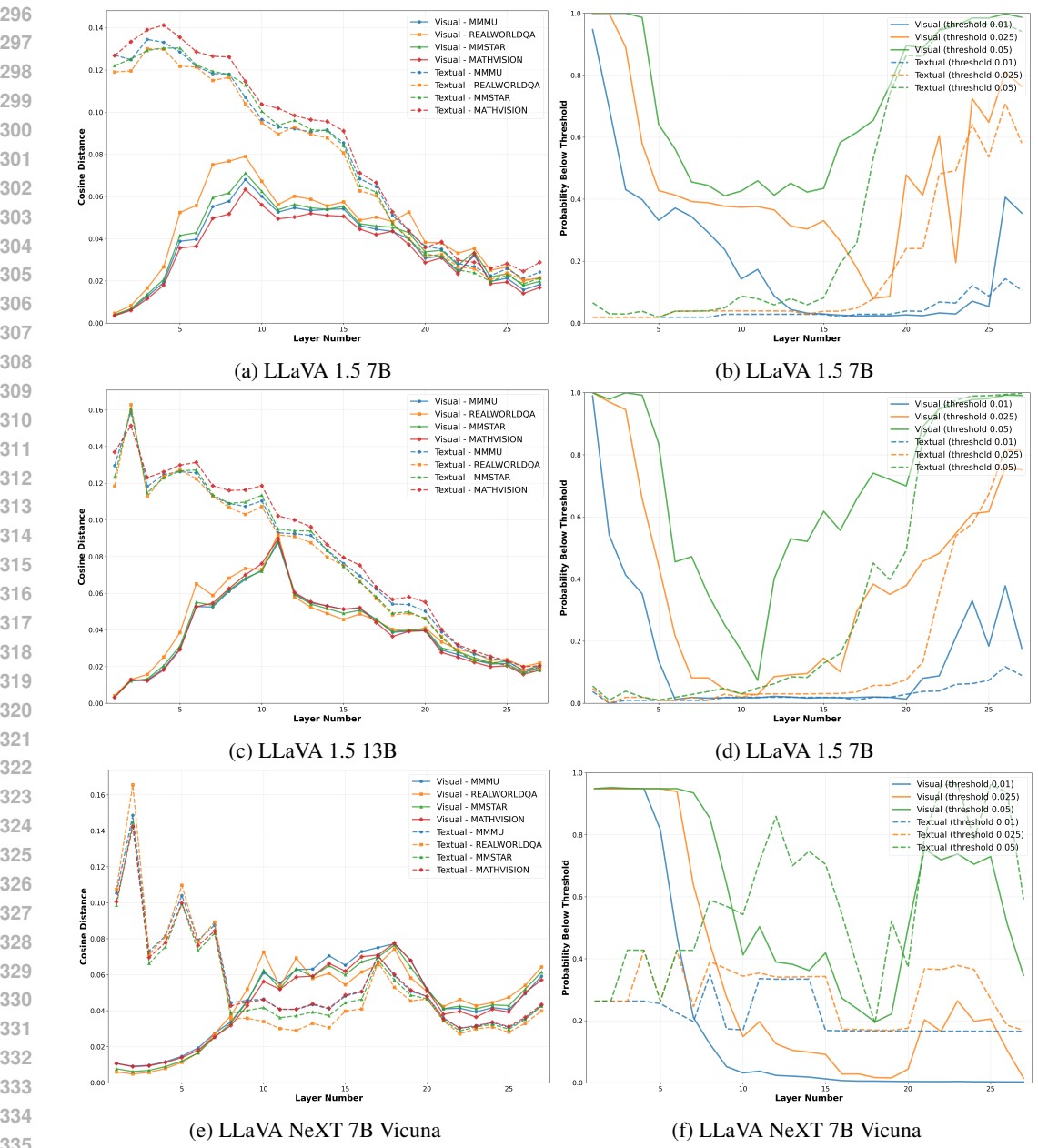

Figure 8: Empirical Geometric Redundancy and Proximal Redundancy between hidden states versus layer. Across all the Multimodal Reasoning task (see Table 3) on LLaVA 1.5/1.6, the early and late layer vision tokens have low adjacent token cosine distances.

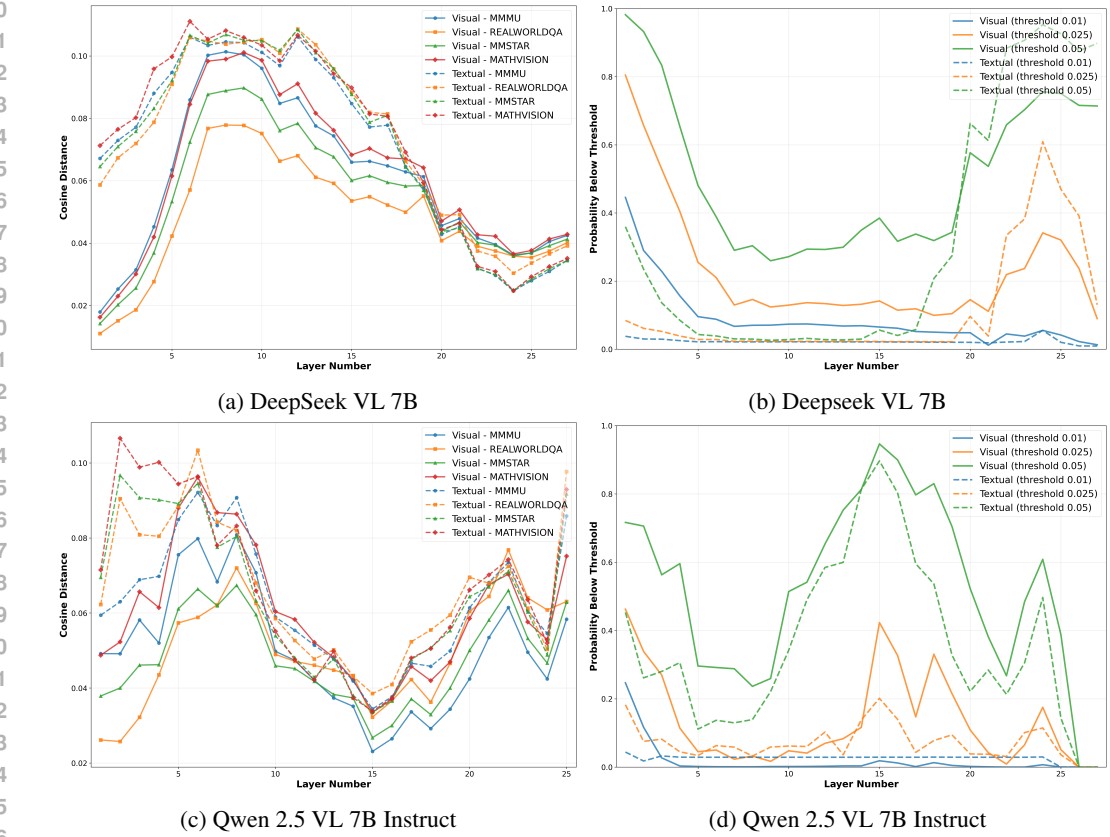

(a) DeepSeek VL 7B

(b) Deepseek VL 7B

(c) Qwen 2.5 VL 7B Instruct

(d) Qwen 2.5 VL 7B Instruct

Figure 9: Empirical Geometric and Proximal Redundancy versus layer. Across the Multimodal Reasoning task (see Table 3) on Qwen 2.5 VL Instruct and DeepSeek VL 7B, the early and late layer vision tokens have low adjacent token cosine distances.

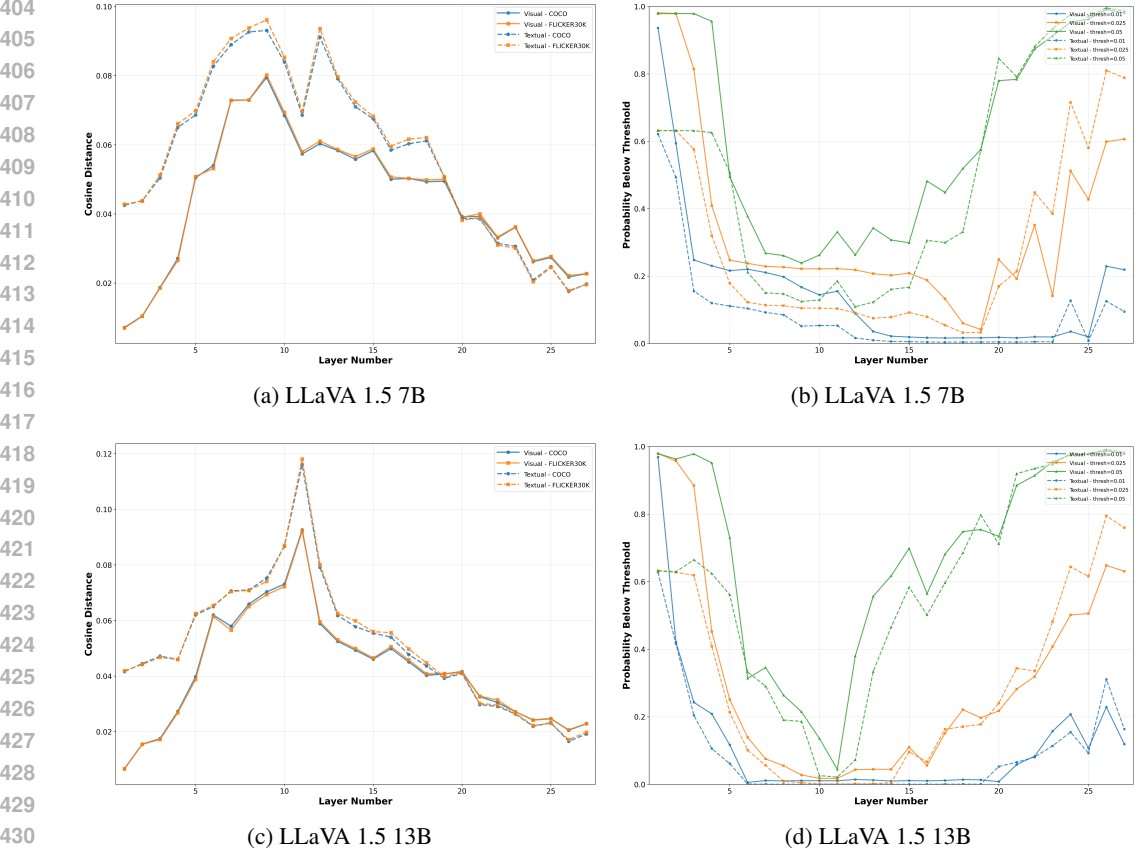

(a) LLaVA 1.5 7B

(b) LLaVA 1.5 7B

(c) LLaVA 1.5 13B

(d) LLaVA 1.5 13B

Figure 10: Empirical Geometric and Proximal Redundancy versus layer on LLaVA 1.5 architectures. Across the Captioning task (see Table 3) on LLaVA 1.5 7B and 13B, the early and late layer vision tokens have low adjacent token cosine distances.

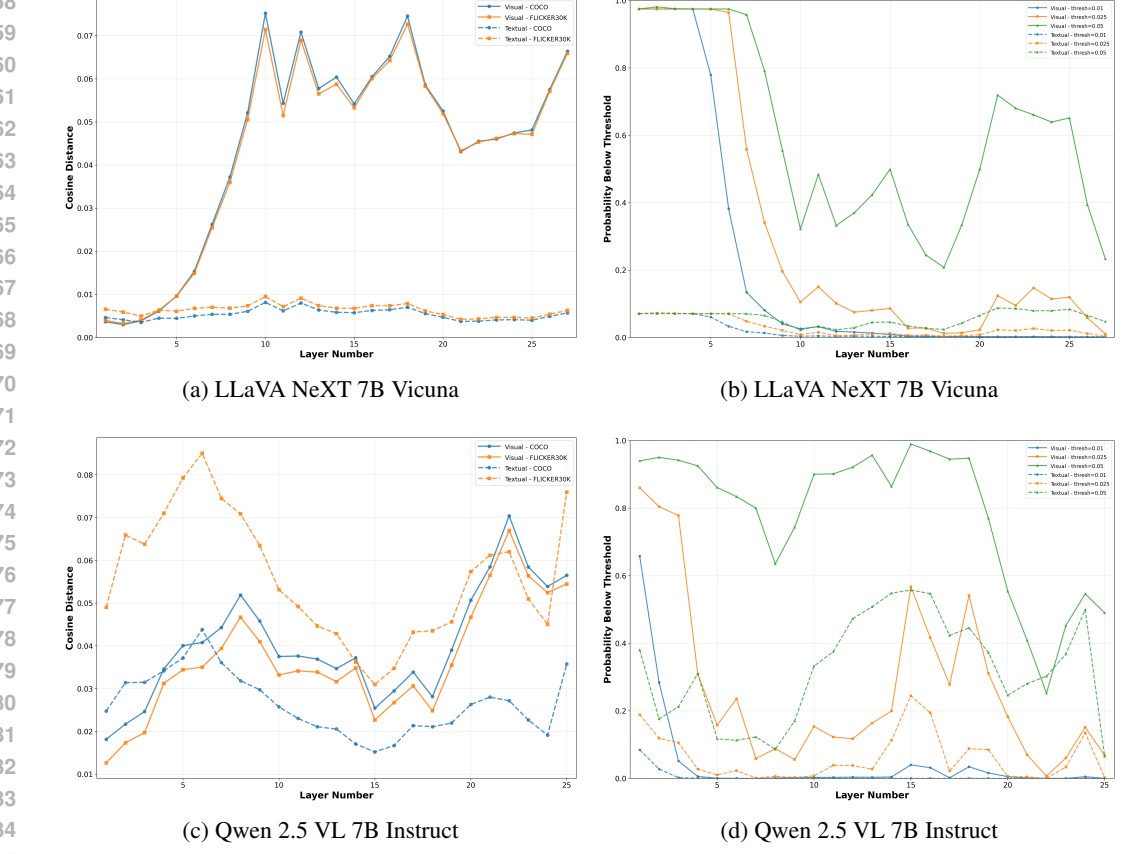

(a) LLaVA NeXT 7B Vicuna

(b) LLaVA NeXT 7B Vicuna

(c) Qwen 2.5 VL 7B Instruct

(d) Qwen 2.5 VL 7B Instruct

Figure 11: Empirical Geometric and Proximal Redundancy versus layer. Across the Captioning task (see Table 3) on LLaVA NeXT 7B Vicuna and Qwen 2.5 VL Instruct, the early and late layer vision tokens have low adjacent token cosine distances.

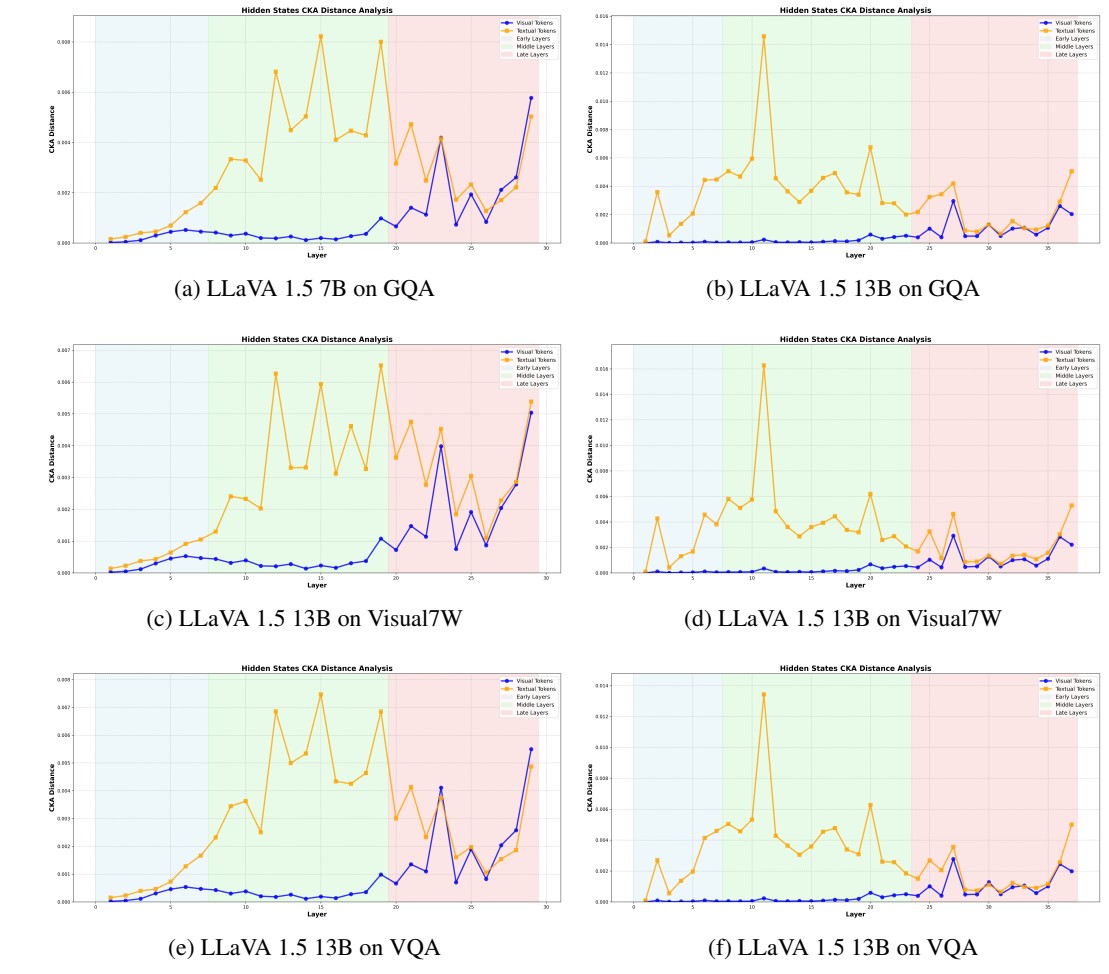

(a) LLaVA 1.5 7B on GQA

(b) LLaVA 1.5 13B on GQA

(c) LLaVA 1.5 13B on Visual7W

(d) LLaVA 1.5 13B on Visual7W

(e) LLaVA 1.5 13B on VQA

(f) LLaVA 1.5 13B on VQA

Figure 12: CKA distance (Kornblith et al., 2019) on LLaVA 1.5 architectures. The early vision tokens seem to have low adjacent token CKA distances.

## D   FURTHER RESULTS FROM SECTION 4.2

In this section, we include plots of the skipping experiments for LLaVA 1.5 7B/13B and LLaVA NeXT 7B Vicuna on all datasets.

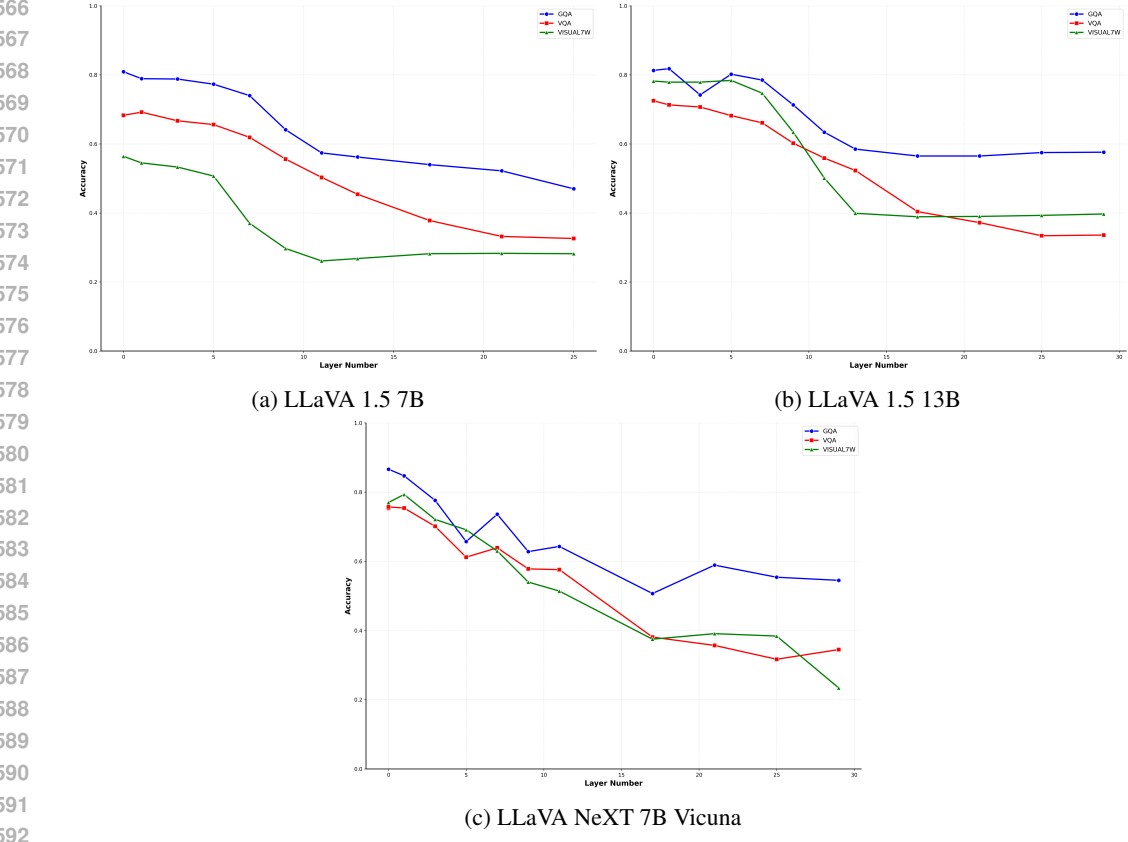

(a) LLaVA 1.5 7B

(b) LLaVA 1.5 13B

(c) LLaVA NeXT 7B Vicuna

Figure 13: Skipping model accuracy versus layer. Run across all of the General VQA tasks (see Table 3) on the LlaVA models. The sharpest decrease in the early-middle layers.

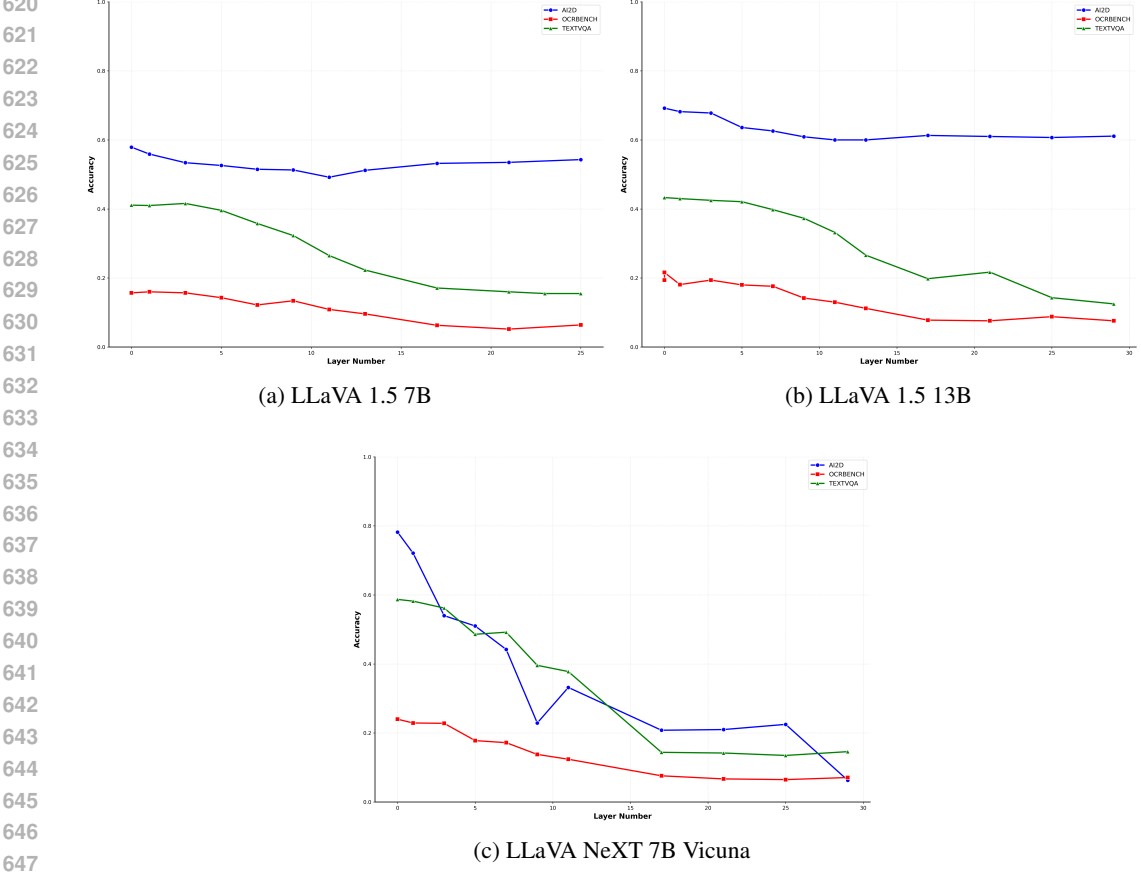

(a) LLaVA 1.5 7B  (b) LLaVA 1.5 13B

(c) LLaVA NeXT 7B Vicuna

Figure 14: Skipping model accuracy versus layer. Run across all of the Text/Doc VQA tasks (see Table 3) on the LLaVA models. The sharpest decrease in the early-middle layers.

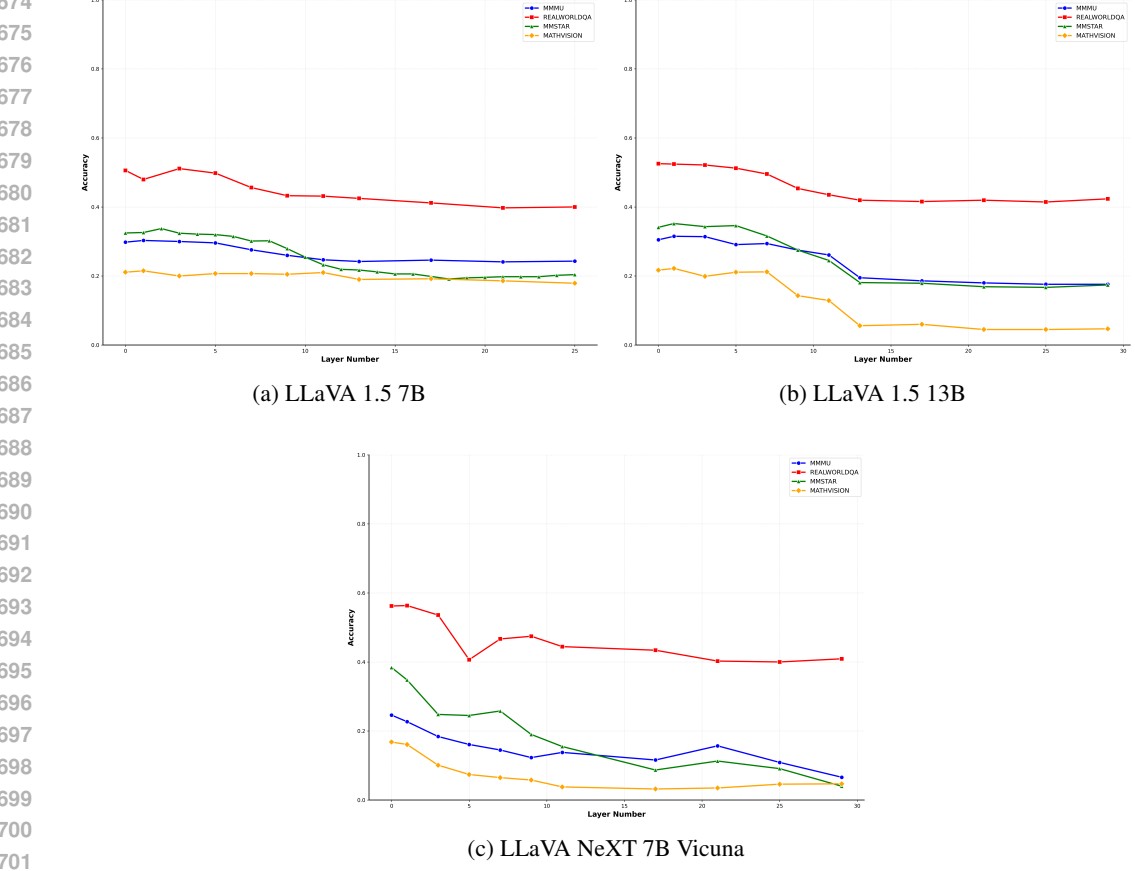

(a) LLaVA 1.5 7B  (b) LLaVA 1.5 13B

(c) LLaVA NeXT 7B Vicuna

Figure 15: Skipping model accuracy versus layer. Run across all of the Multi-modal reasoning tasks (see Table 3) on the LLaVA models. The sharpest decrease in the early-middle layers.

## E  CONNECTION TO PID

For readers familiar with information theory, much of the vocabulary and intuition discussed regarding informational redundancy may sound very similar to notions of redundant and unique information in partial information decomposition (PID). PID (of 3 finite-support random variables) proposes that the mutual information $I(X; Y, Z)$ can be decomposed into the following terms.

1. Unique Information: $Uni(X : Y \backslash Z)$ and $Uni(X : Z \backslash Y)$ for the unique information that $Y$ contains about $X$ and $Z$ contains about $X$ respectively.

2. Redundant Information: $Red(X : Y, Z)$, which is the information about $X$ that both $Y$ and $Z$ share.

3. Synergistic Information (sometimes called Shared Information): $Syn(X : Y, Z)$, which is the information about $X$ that can only be derived from the combination of both $Y$ and $Z$.

The proposed decomposition of the mutual information is given by the following definition.

**Definition 5** (Partial Information Decomposition)**.**

$$I(X; Y, Z) \triangleq Uni(X : Y \backslash Z) + Uni(X : Z \backslash Y) + Red(X : Y, Z) + Syn(X : Y, Z)$$

$$I(X; Y) \triangleq Uni(X : Y \backslash Z) + Red(X : Y, Z)$$

$$I(X; Z) \triangleq Uni(X : Z \backslash Y) + RED(X : Y, Z).$$

In fact, the connection between PID and informational redundancy can be somewhat formalized through the following observation.

**Lemma 5.** *Let $X, Y$ be discrete random variables. Then $Uni(X : X \backslash Y) = H(X|Y)$.*

*Proof.* From the chain rule for mutual information we know that $I(X; Y|Z) = I(X; Y, Z) - I(X; Z)$. Using Definition 5 we see that $I(X; Y|Z) = Uni(X : Y \backslash Z) + Syn(X : Y, Z)$ and similarly, $I(X; Z|Y) = Uni(X : Z \backslash Y) + Syn(X : Y, Z)$. Thus $I(X; Y|X) = 0 = Uni(X : Y \backslash X) + Syn(X : X, Y)$. By the non-negativity of PID we get that $Uni(X : Y \backslash X) = Syn(X : X, Y) = 0$. Thus, $I(X; X|Y) = H(X|Y) = Uni(X : X \backslash Y)$. $\square$

Thus, if one considers $X = X_\ell, Y = X_{\ell-1}$, we recover our definition of informational redundancy using PID. This also offers a PID interpretation of redundancy: the unique information about the current layer, which only the current layer has, should be low. Further, since PID considers three random variables, this also allows us to consider a combination of our functional and informational redundancy by considering the quantity $Uni(Z : X_\ell \backslash X_{\ell-1})$. This would be the unique information that $X_\ell$ has about a target random variable $Z$ that $X_{\ell-1}$ does not have.

The unique information quantity $Uni(X : Y \backslash Z)$ also has a widely accepted definition given by Bertschinger et al. (2014), which is the solution to following convex optimzation problem:

**Definition 6** (BROJA definition; (Bertschinger et al., 2014))**.**

$$Uni(X : Y \backslash Z) \triangleq \min_{Q \in \Delta_P} I_Q(X : Y|Z) \tag{41}$$

*where $\Delta$ is the set of all joint distributions on $X, Y, Z$ and $\Delta_P = \{Q \in \Delta : Q(X = x, Y = y) = P(X = x, Y = y) \text{ and } Q(X = x, Z = z) = P(X = x, Z = z) \ \forall x \in supp(X), y \in supp(Y), z \in supp(Z)\}$. That is the set of distributions that agree on the marginals.*

If one can bound this value, then they recover a "functional information-theoretic redundancy".

