# OpenReview forum: "Skip-It? Theoretical Conditions for Layer Skipping in Vision–Language Models"
_ICLR.cc/2026/Conference — Submitted to ICLR 2026_

### Official Review · Reviewer_Ttaf · 2025-10-27

**Soundness:** 2
**Presentation:** 2
**Contribution:** 2
**Rating:** 4
**Confidence:** 4

**Summary:**

The paper shows that high cosine similarity between adjacent VLM layers indicates where layer skipping can improve inference speed without hurting performance. Experiments confirm early vision layers and late layers have high redundancy and low cross-attention, allowing safe skipping for speedup. However, the theoretical proofs rely on invalid assumptions that don't apply to transformers.

**Strengths:**

- Establishes formal connections between measurable redundancy metrics (cosine distance) and task-relevant redundancy, providing principled guidance for layer skipping.
- Tests across multiple VLMs and diverse tasks, showing consistent redundancy patterns in early vision and late layers.

**Weaknesses:**

- The theoretical framework relies on assumptions that don't hold for actual transformer-based VLMs, particularly the Markov chain assumption (X_{ℓ-1} → X_ℓ → Z) which ignores residual connections.
- The identified redundancy patterns (high redundancy in early vision layers and late layers) align with known transformer behavior and are not particularly surprising.
- The claim that "both high redundancy and small cross-attention are necessary conditions" for safe layer skipping is presented without formal proof, relying only on empirical validation.

**Questions:**

1. Can you provide empirical measurements of the Lipschitz constants (α, β) for the function h(x,y) in actual VLMs? Without these values, how can we interpret the bounds in Theorems 1-2?
2. The Markov assumption X_{ℓ-1} → X_ℓ → Z appears violated in transformers due to residual connections and attention across all layers. Could you either (a) empirically test this assumption, or (b) reformulate your results without it?
3. Your theorems require discrete/finite support for hidden states (Theorems 3-5), but representations in each layer are continuous. What discretization scheme do you propose, and how does discretization error affect your bounds?
4. Have you tested whether your redundancy patterns hold for other tasks (e.g., image captioning) rather than just discriminative VQA tasks? The information flow might differ significantly.
5. Could your redundancy analysis guide other efficiency methods (e.g., which layers to keep during distillation) beyond just layer skipping? This might increase the paper's impact.

---

> ### Author Response · Authors · 2025-11-22
> **Response to Reviewer Ttaf**
>
> We thank the reviewer for noting the formal connections established in the paper and our thorough experiments under diverse settings. We address their comments and questions below.
>
> **Weaknesses/Questions**:
> ```
> The theoretical framework relies on assumptions.......... particularly the Markov chain assumption (X_{ℓ-1} → X_ℓ → Z) which ignores residual connections.
> ```
> ```
> The Markov assumption X_{ℓ-1} → X_ℓ → Z appears violated........Could you either (a) empirically test this assumption, or (b) reformulate your results without it?
> ```
> We would like to respectfully disagree with the reviewer; since the hidden states used in our paper are taken after the residual has been added within the decoder layer, the Markovity assumption holds. We do thank the reviewer for bringing up this potential confusion, and we add this clarification to the paper to make it clearer.
> ```
> The identified redundancy patterns (high redundancy in early vision layers and late layers) align with known transformer behavior......
> ```
> While redundancy in the early/late layers is widely acknowledged, our work provides one of the first theoretical groundings for these empirical redundancies as well as allows to predict when such redundancy may be feasibly exploited, without requiring downstream performance measures (unlike the layer skipping methods that need hyperparameter tuning).
> ```
> The claim that "both high redundancy and small cross-attention are necessary conditions" for safe layer skipping is presented without formal proof......
> ```
> We thank the reviewer for pointing this oversight. As pointed out by reviewer b2f6, we acknowledge that it may not be necessary to have low cross attention for redundancy. Nevertheless, we now formally establish that it is indeed a **sufficient** condition (rather than necessary and sufficient) where minimal attention between the modalities and high redundancy does allow for late entry without performance degradation. A formal statement is presented in Theorem 6 of the updated manuscript.
>
> ```
> Empirical measurements of the Lipschitz constants............how can we interpret the bounds in Theorems 1-2?
> ```
> The use of Lipschitz assumptions is consistent with other work in this area [1]. We still can, nevertheless, obtain an upper bound on the Lipschitz constant by analyzing the spectral norm of the layers, which we can approximate as the spectral norm of the output matrix times the value matrix. When computing this spectral norm for layer 4, we get ~5.89. It is of note that the exact value of this Lipschitz constant is of less importance than the overall trend that functional redundancy decays linearly as geometric redundancy, so our empirical notions of redundancy translate to more informative notions.
>
> [1] Skip-Vision: Efficient and Scalable Acceleration of Vision-Language Models via Adaptive Token Skipping: Zeng et al, ICCV 2025
> ```
> ...Finite support for hidden states (Theorems 3-5)...... What discretization scheme do you propose.......?
> ```
> We thank the reviewer for pointing out this limitation. We have extended the information-theoretic results to hold for the continuous case. We are also currently working on including a quantization approach where we uniformly quantize the random variables and then use the discrete entropy along with an error term from the quantization bins to find an upper bound with some error. We will try to include this within the rebuttal period.
> ```
> .......redundancy patterns hold for other tasks rather than just discriminative VQA tasks?......
> ```
> At the reviewer's request, we add tests on COCO and Flickr30k Captioning datasets across all models. We found that there is more redundancy in the textual tokens in the early layers of each model, compared to the VQA task, which had minimal textual redundancy in the early layers. The early layer vision tokens had a sharper increase in distance, indicating these tokens in early layers became non-redundant faster. Both vision and textual tokens show redundancy in the later layers, across tasks. These new results can be seen in Figures 10/11.
> ```
> Could your redundancy analysis guide other efficiency methods (e.g., which layers to keep during distillation) beyond just layer skipping? This might increase the paper's impact.
> ```
> Yes, our redundancy analysis is general and could allow one to determine layers to keep during distillation. Our framework does not require an estimate of downstream performance to determine redundancy. Thus, it allows for the user to identify redundant layers in a model when deploying to an entirely new domain. Our study on late entry and early exit confirm this as our redundancy analysis allows us to predict which layers to skip on many tasks, without needing labeled data to measure downstream performance.
>
> With this, we hope to have addressed the main concerns of the reviewer to a satisfactory extent and that our comments have improved their opinion of our work. We remain on standby for any further questions.

---

> > ### Author Response · Authors · 2025-12-04
> > **Summary of Response to Reviewer Ttaf**
> >
> > We respectfully disagree with the reviewer that the Markovity assumptions do not hold because the hidden states are taken after the residuals are added thus ensuring that Markovity does in fact hold. We also note that while the redundancy in early and late layers is a known phenomena, our paper is the first to provide theoretical groundings for the empirical redundancies discussed in previous works. We have now also added a theorem proving the previous assertion that high visual redundancy and small inter-modal attention is sufficient for layer skipping. We note that a trivial bound on our Lipschitz constants can be found by analyzing the spectral norm of the layers. We also have modified our information-theoretic analyses to account for continuous random variables rather than discrete. Finally, we have added additional results on image captioning datasets and results can be found in Figures 10/11. We also note that our redundancy analysis could guide other efficiency methods and a section detailed such has been added to the paper. Because our analysis does not require an esimate of downstream performance it could allow for a user to identify redundant layers in a model even when deploying in a new domain.

---

### Official Review · Reviewer_b2f6 · 2025-11-01

**Soundness:** 2
**Presentation:** 3
**Contribution:** 2
**Rating:** 4
**Confidence:** 3

**Summary:**

The paper proposes a principled framework for determining when layer skipping in vision–language models (VLMs) improves efficiency without degrading accuracy. The central idea is to relate easy-to-measure, adjacent-layer similarity to stronger notions of redundancy that are sufficient for safe skipping. Concretely, the authors define geometric and proximal redundancy based on cosine distance and tail probabilities between consecutive hidden states, and show these imply functional redundancy and are connected to informational redundancy via Fano-type bounds and a lower bound on mutual information upper-bounds performance differences between optimal predictors from layers. The framework is coupled with a cross-modality condition: low cross-attention in a layer band is required so that skipping a modality does not harm the other. Empirically, the paper measures per-layer average cosine distance and proximity probabilities, and visualizes VAR over layers. On LLaVA-1.5 (7B/13B) and LLaVA-NeXT (7B), early and late layers show high redundancy and low VAR, while mid layers concentrate cross-modal fusion. Layer-skipping experiments corroborate the theory: when redundancy and low VAR hold, late entry or early exit  preserves accuracy; violating the conditions degrades performance markedly.

**Strengths:**

1. Originality. The paper unifies four redundancy notions into a single implication chain and turns them into practical, theory-backed criteria for when late entry and early exit should work in VLMs.
2. Theory Quality. The analysis is sound and shows that small changes between adjacent layers imply small performance differences, and that low conditional uncertainty connects information- and function-level redundancy; combining this with low cross-modal attention yields clear skip conditions.
3. Experiments Quality. Measurements of adjacent-layer similarity and cross-attention match the theory: early/late layers are redundant, mid layers do fusion, and skipping only works reliably where the conditions hold while violating them causes clear drops.
4. Clarity & Significance. The presentation is clean and the framework can standardize safe-skipping decisions, explaining why prior heuristics often work and offering a pathway to reduce inference cost without retraining.

**Weaknesses:**

1. Metric range consistency. The cosine-distance range is used inconsistently; normalizing the distance or generalizing the bound would resolve this.
2. Discrete vs. continuous assumptions. Information-theoretic bounds assume finite supports while hidden states are continuous; a simple quantization scheme and error accounting would close the gap.
3. Markov and Lipschitz assumptions. The required independence and smoothness conditions may be violated in modern architectures; approximate versions and empirical checks would improve realism.
4. Cross-modal dependence via attention only. Using attention ratios alone doesn’t guarantee causal importance; adding attribution or ablation tests would strengthen the claim.
5. Compute reporting. Claimed speedups aren’t backed by latency, throughput, memory, or energy measurements; reporting these under fixed settings would quantify real gains.
6. Comparative baselines. There’s no head-to-head with matched-compute baselines or popular alternatives; adding these would clarify benefits.

**Questions:**

1. How are continuous hidden states discretized for the information-theoretic bounds, and how sensitive are decisions to the quantization level?
2. What end-to-end speedups, memory savings, and potential KV-cache overheads do you observe on common GPUs and batch sizes?
3. Do the skip decisions remain consistent when using alternative similarity measures (e.g., CKA/PWCCA/SVCCA) instead of cosine distance?
4. Please refer to the section on Weaknesses for other questions.

---

> ### Author Response · Authors · 2025-11-22
> **Response to Reviewer b2f6**
>
> We thank the reviewer for highlighting our unified framework and the quality of our theory and experiments. We respond to their comments and questions below.
>
> **Weaknesses/Questions**:
> ```
> Metric range consistency.
> ```
> We thank the reviewer for pointing this out. The informational redundancy notions now assume unit-normalized vectors to agree with the functional redundancy notions. The theoretical results do not change with this assumption.
> ```
> Discrete vs. continuous assumptions. Information-theoretic bounds assume finite supports while hidden states are continuous; a simple quantization scheme and error accounting would close the gap.
> ```
> ```
> How are continuous hidden states discretized for the information-theoretic bounds, and how sensitive are decisions to the quantization level?
> ```
> As stated in the common response, we have expanded our theoretical results to the continuous case. The results remain similar under some additional assumptions on entropy being bounded from below. We are currently working on including a quantization approach where we uniformly quantize the random variables and then use the discrete entropy with an error term from the quantization bins to find an upper bound with some error. We hope to include this within the rebuttal period.
> ```
> Markov and Lipschitz assumptions. The required independence and smoothness conditions may be violated in modern architectures; approximate versions and empirical checks would improve realism.
> ```
> The hidden states we used for analysis are taken after the residual is added so the Markovity assumption holds. This justification has been added to our paper to make this clearer. The use of Lipschitz boundedness is consistent with prior work in this area [1]. For completeness, one can obtain that a trivial upper bound on the Lipschitz constant via the spectral norms of the self-attention/feed-forward network. Since the Lipschitz constant of softmax is upper bounded by one, we can approximate an upper bound on the Lipschitz constant of the attention in a layer by taking the spectral norm of the output matrix times the value matrix. For example, for layer 4, we find the Lipschitz constant to be ~5.89.
>
> [1] Skip-Vision: Efficient and Scalable Acceleration of Vision-Language Models via Adaptive Token Skipping: Zeng et al., ICCV 2025
> ```
> Cross-modal dependence via attention only. Using attention ratios alone doesn’t guarantee causal importance; adding attribution or ablation tests would strengthen the claim.
> ```
> This is a very astute observation, and we thank the reviewer for pointing out this oversight. We acknowledge that a high attention-ratio between vision and text does not mean that there is meaningful information communicated between the two modalities. Therefore, it may not be necessary to have low cross attention for redundancy. Nevertheless, we now formally establish that it is indeed a **sufficient** condition (rather than necessary and sufficient) where minimal attention between the modalities and high redundancy does allow for late entry without performance degradation. A formal proof is presented in Theorem 6 of the updated manuscript.
> ```
> Compute reporting. Claimed speedups aren’t backed by latency, throughput, memory, or energy measurements;
> ```
> ```
> Comparative baselines. There’s no head-to-head with matched-compute baselines or popular alternatives; adding these would clarify benefits.
> ```
> ```
> What end-to-end speedups, memory savings, and potential KV-cache overheads do you observe on common GPUs and batch sizes?
> ```
> We stress again that our paper does not propose a new efficiency method but rather provide analytical conditions for when layer skipping or other pruning techniques are feasible. For insights into speedups, memory improvements etc., the reviewer may refer directly to the cited works, since we follow their implementations. Rather than a competing methodology, our core contribution is complementary to the aforementioned methods in that, while they rely on downstream performance to adaptively tune hyperparameters for when to skip, we can (provably) predict redundancy to an extent without any downstream labels, by simply analyzing the evolution of representations through the layers.
> ```
> Do the skip decisions remain consistent when using alternative similarity measures (e.g., CKA/PWCCA/SVCCA) instead of cosine distance?
> ```
> We have included preliminary results using an average CKA distance of adjacent layer representations in Appendix C (Figure 12). In general, the CKA distance (1-CKA similarity) is very low for vision tokens in the early layers but is high in the later layers. As such, the skip decision remains consistent for early layers but diverges from our cosine-distance based approach for the late layers.
>
> With this, we hope to have addressed the main concerns of the reviewer to a satisfactory extent and that our comments have improved their opinion of our work. We remain on standby for any further questions.

---

> > ### Author Response · Authors · 2025-12-04
> > **Summary of Response to Reviewer b2f6**
> >
> > As stated in the common response, this paper does not claim to provide a new VLM efficiency method but rather provides a theoretical framework for analyzing redundancy in VLMs which we then use to analyze an existing layer-skipping method. The paper has edited to make this point clearer. We have also modified our information-theoretic analyses to account for continuous random variables rather than discrete. We note that the Markov assumptions we make do hold for our layer-skipping framework as the hidden states are taken after the residual calculation ensuring that Markovity holds. We have also included a theorem to formally prove the previous assertion made that high redundancy and minimal inter-modal attention is sufficient for layer-skipping. Finally, we have included preliminary results on average CKA distance (instead of cosine distance) in Appendix C.

---

### Official Review · Reviewer_7T6v · 2025-11-01

**Soundness:** 2
**Presentation:** 2
**Contribution:** 2
**Rating:** 4
**Confidence:** 4

**Summary:**

This paper introduces a novel visual token pruning method for VLMs, designed to enhance their inference efficiency. The core methodology is operationally simple, involving the skipping of visual tokens at specific layers, such as implementing strategies the authors refer to as 'Late Entry' and 'Early Exit'. The authors place particular emphasis on their analytical framework, which is proposed to determine which layers can be safely skipped. This analysis relies primarily on data derived from cross-attention scores. A key highlight of this work is its demonstration, via the proposed methodology, that visual tokens in the shallow layers of the VLM are similarly dispensable to those in the deep layers. While the latter (the redundancy of deep-layer tokens) is a well-established community consensus, the former (the non-essential nature of shallow-layer tokens) has not been widely discussed. Therefore, the conclusions drawn from this work are of significant inspirational value for the VLM research community.

**Strengths:**

1.  The paper provides a relatively comprehensive theoretical analysis, establishing a solid justification for determining which layers' tokens can be skipped.
2.  The discovery that visual tokens in most shallow layers can also be skipped is a novel insight that is quite insightful for the VLM research community.

**Weaknesses:**

1.  **Insufficient Experimental Validation:** For a paper proposing a VLM efficiency optimization method, the experimental validation is rather incomplete. It lacks direct comparisons in terms of both performance and efficiency against other similar methods (e.g., PDrop, TwigVLM et al.).
2.  **Clarity of Results:** The presentation of the final results is unclear. The paper fails to report definitive performance metrics on specific downstream tasks, making it difficult to assess the practical benefits.
3.  **Writing and Terminology:** The paper's clarity needs significant improvement. For instance, the terms "guessing" and "not guessing" in Table 2 are ambiguous, and their intended meanings are not clearly defined. Additionally, the initial mention of "cross-attention" requires a more precise definition in this context. As written, it risks creating confusion with modifications to traditional "cross-attention layers" which hinders reader comprehension.

**Questions:**

1.  Is the layer-skipping strategy uniform across different tasks, or is it adapted based on the task?
2.  How does the proposed method compare in both performance (accuracy) and efficiency (e.g., latency, throughput) against other SOTA VLM acceleration strategies?
3.  The experiments are conducted on LLaVA-1.5, which may be considered undertrained or outdated. Do the paper's conclusions hold for more recent, powerful models (e.g., Qwen2.5-VL, Gemma3-VL, Qwen3-VL) and across a broader spectrum of tasks?

---

> ### Author Response · Authors · 2025-11-22
> **Response to Reviewer 7T6v**
>
> We thank the reviewer for noting our comprehensive theoretical analysis and novel insight. Below, we address their comments and concerns.
>
> **Weaknesses/Questions**
> ```
> Insufficient Experimental Validation: For a paper proposing a VLM efficiency optimization method, the experimental validation is rather incomplete. It lacks direct comparisons in terms of both performance and efficiency against other similar methods (e.g., PDrop, TwigVLM et al.).
> ```
> ```
> Clarity of Results: The presentation of the final results is unclear. The paper fails to report definitive performance metrics on specific downstream tasks, making it difficult to assess the practical benefits.
> ```
> To clarify, we do not propose a new efficiency method. Rather, we aim to present a framework that can be used to identify and describe the redundancy in VLMs. Based on the posited conditions, we can demonstrably identify both the scope for early vision exit (as exploited in methods such as Visual Tokens Withdrawal (VTW)) and late vision entry (such as DeepInsert) as two classes of pruning techniques to study as a case study. The token pruning aspect (i.e. adaptively selecting a subset of important tokens from the long vision context) is complementary to this identified redundancy, and therefore, can be combined with both early exit and late entry.
>
> To better reflect this fact, we have updated the paper to more explicitly describe the aim of this work. We reorganized the paper to first propose the redundancy framework (Section 3), and then we explain how early exit and late entry are used as our case study to analyze the redundancy of vision and text tokens (Section 4). We then apply our analysis to both early exit and late entry vision token skipping to see how skipping redundant tokens can be done without model degradation.
>
> ```
> Writing and Terminology: The paper's clarity needs significant improvement. For instance, the terms "guessing" and "not guessing" in Table 2 are ambiguous, and their intended meanings are not clearly defined. Additionally, the initial mention of "cross-attention" requires a more precise definition in this context.
> ```
> Guessing and not guessing in table 2 referred to the MMMU and MathVision datasets, which achieve performance similar to guessing on all models we experimented on. We addressed this point in the captions of tables 1 and 2. We thank the reviewer for pointing out this ambiguity in our statement of "cross-attention". Here, we refer to "cross-attention" as the inter-modality attention. We have adjusted our terminology throughout the paper.
>
> ```
> Is the layer-skipping strategy uniform across different tasks, or is it adapted based on the task?
> ```
> This question raised a good point about how redundancy depends on the model and task. Based on our results in section 4 (table 1), we found that the redundant layers (which can be skipped) are similar across the VQA and Multimodal reasoning tasks but differ for Captioning. We found that the specific layers which can be skipped are model dependent.
> ```
> How does the proposed method compare in both performance (accuracy) and efficiency (e.g., latency, throughput) against other SOTA VLM acceleration strategies?
> ```
> We do not claim to contribute a pruning method. We propose a framework which can be used to theoretically characterize when pruning methods such as DeepInsert, FastV, and VTW are feasible to improve efficiency without model degradation. These methods work due to the same underlying model mechanisms, so we aim to provide a grounded framework to begin explaining why they work. From a practical point of view, our work can be used to improve all of these methods by indicating when they can be used.
> ```
> The experiments are conducted on LLaVA-1.5, which may be considered undertrained or outdated. Do the paper's conclusions hold for more recent, powerful models (e.g., Qwen2.5-VL, Gemma3-VL, Qwen3-VL) and across a broader spectrum of tasks?
> ```
> As per the reviewer’s request, we expanded our results to Qwen2.5-VL. We found that vision late entry in Qwen2.5-VL significantly degrades performance by layer 4 compared to LlaVA 1.5/1.6, which sees similar degradation later, across all tasks. For early exit, Qwen2.5-VL degrades when early exiting at layer 24 in multi-modal reasoning and VQA tasks and at layer 20 for Captioning. These results are in line with what we expected from computing redundancy of these models. In general, adding the Qwen2.5-VL model helped us strengthen our claim that early exit and late entry are largely model dependent. Additionally, we included the Image Captioning Task, using the COCO and Flickr30k datasets to expand upon the tasks in which we test our framework.
>
> With this, we hope to have addressed the main concerns of the reviewer to a satisfactory extent and that our comments have improved their opinion of our work. We remain on standby for any further questions.

---

> ### Author Response · Authors · 2025-12-04
> **Summary of Response to Reviewer 7T6v**
>
> As stated in the common response, this paper does not claim to propose a new VLM efficiency method but rather proposes a theoretical framework to analyze redundancy in VLMs and then uses that framework to justify existing layer-skipping techniques. Our paper has been modified to make this fact clearer and explain some unclear terminology. We have also extended our experimentation to include Qwen2.5VL as requested.

---

### Author Response · Authors · 2025-11-22
**Common Response to the reviewers**

We wanted to thank each reviewer for their insights and helpful comments. In this section, we provide more details into common comments and concerns among reviewers.

**Scope of contribution (Reviewers 7T6v, b2f6):** Based on common themes across the reviews, we want to clearly restate our core contribution. The goal of this paper is to provide a framework which describes redundancy of tokens at different layers. While previous layer skipping and token pruning methods (including DeepInsert, VTW, FastV,TwigVLM, etc.) leverage this redundancy empirically, a formal treatment has been lacking. Our framework addresses this gap by grounding the notion of redundancy theoretically and applying it to analyze early exit and late entry pruning as a unified class of techniques that exploit layer redundancy. Therefore, our contribution is complementary to these works, in that, while these methods rely on downstream performance to adaptively tune hyperparameters for when to skip, we can (provably) predict redundancy to an extent without any downstream labels, by simply analyzing the evolution of representations through the layers and using our stated conditions. We have revised the paper to clearly articulate this.

**Experimental breadth (Reviewers 7T6v, b2f6, Ttaf):** We have substantially extended our experimental evaluation. Specifically, we: (1) added experiments on Qwen 2.5 VL, revealing that redundancy patterns are model-dependent; (2) included image captioning tasks (COCO, Flickr30k), demonstrating that vision token redundancy is task-dependent in later layers; and (3) evaluated redundancy using Centered Kernel Alignment (CKA), showing that our findings are robust to the choice of semantic distance metric. We are also conducting experiments on Qwen 3 VL and welcome any additional suggestions.

**Theory change (Reviewers b2f6, Ttaf):** We address several theoretical concerns raised by the reviewers:
- Markovity assumption (Theorem 5): We want to clarify that since hidden states are retrieved after the residual connection, the Markov property holds on the post-residual representations. We have clarified this in the revised paper.
- Lipschitz assumptions: These assumptions are consistent with prior work of this ilk [1]. Nevertheless, for completeness, we also compute empirical upper bounds via spectral norms (e.g., ~5.89 for layer 4). Note that while these can be tightened further using bound propagation [2] and other related techniques, our empirical contributions are not reliant on them.
- Discrete vs. continuous entropy: We have revised the information-theoretic results to use continuous random variables and differential entropy. This requires an additional boundedness assumption on differential entropy in Theorem 5. We are also developing a uniform quantization approach to recover discrete entropy bounds, which we aim to include during the rebuttal period.
- We also added formal justification for our claim of minimal cross-modal attention from vision to text and high visual redundancy being sufficient conditions for the late entry of vision tokens. This is established in Theorem 6, and a formal proof is provided in the Appendix.

[1] Skip-Vision: Efficient and Scalable Acceleration of Vision-Language Models via Adaptive Token Skipping: Zeng et al., ICCV 2025

[2] Efficiently Computing Local Lipschitz Constants of Neural Networks via Bound Propagation: Shi et al., 2022

---

### Meta-Review · Area_Chair_tSxE · 2025-12-15

**Summary:**

The paper proposes a novel framework for evaluating layer redundancy in VLMs using cosine similarity and a probability-based notion of redundancy. The reviewers’ main concerns are about the limited experimental evaluation and several unvalidated theoretical assumptions. While a rebuttal was provided, it does not adequately address the primary concern regarding the insufficiency of experimental evidence.

**Reviewer Concerns:**

The rebuttal addressed concerns about unclear terminology and clarified whether the results generalize across different tasks. It partially addressed concerns regarding certain theoretical assumptions. However, key concerns remain outstanding, including the lack of experimental comparisons with other baselines, the absence of reported computational efficiency metrics, and the limited clarity of the experimental results.

**Reviewer Scores:**

Reviewer Ttaf may increase the score, as most of the concerns were addressed in the rebuttal. Reviewers b2f6 and 7t6v are unlikely to do so, as their primary concern regarding the lack of experimental evaluation remains unaddressed.

---

### Decision · Program_Chairs · 2026-01-26

Reject